# The yeast RNA methylation complex consists of conserved yet reconfigured components with m6A-dependent and independent roles

Imke Ensinck[1†], Alexander Maman[2†], Waleed S Albihlal[1‡], Michelangelo Lassandro[3‡], Giulia Salzano[3‡], Theodora Sideri[1], Steven A Howell[1], Enrica Calvani[1], Harshil Patel[1], Guy Bushkin[4], Markus Ralser[1,5], Ambrosius P Snijders[1], Mark Skehel[1], Ana Casañal[3], Schraga Schwartz[2*], Folkert J van Werven[1*]

[1]The Francis Crick Institute, London, United Kingdom; [2]Department of Molecular Genetics, Weizmann Institute of Science, Rehovot, Israel; [3]Human Technopole, Milan, Italy; [4]Whitehead Institute for Biomedical Research, Cambridge, United States; [5]Charité Universitätsmedizin Berlin, Department of Biochemistry, Berlin, Germany

*For correspondence:
schwartz@weizmann.ac.il (SS);
folkert.vanwerven@crick.ac.uk
(FJvW)

[†]These authors contributed
equally to this work
[‡]These authors also contributed
equally to this work

Competing interest: The authors
declare that no competing
interests exist.

Reviewing Editor: Pablo A
Manavella, Universidad Nacional
del Litoral-CONICET, Argentina

**Abstract** *N6*-methyladenosine (m6A), the most abundant mRNA modification, is deposited in mammals/insects/plants by m6A methyltransferase complexes (MTC) comprising a catalytic subunit and at least five additional proteins. The yeast MTC is critical for meiosis and was known to comprise three proteins, of which two were conserved. We uncover three novel MTC components (Kar4/Ygl036w-Vir1/Dyn2). All MTC subunits, except for Dyn2, are essential for m6A deposition and have corresponding mammalian MTC orthologues. Unlike the mammalian bipartite MTC, the yeast MTC is unipartite, yet multifunctional. The mRNA interacting module, comprising Ime4, Mum2, Vir1, and Kar4, exerts the MTC's m6A-independent function, while Slz1 enables the MTC catalytic function in m6A deposition. Both functions are critical for meiotic progression. Kar4 also has a mechanistically separate role from the MTC during mating. The yeast MTC constituents play distinguishable m6A-dependent, MTC-dependent, and MTC-independent functions, highlighting their complexity and paving the path towards dissecting multi-layered MTC functions in mammals.

## eLife assessment

This **fundamental** study identifies the components of the N6-methyladenosine methyltransferase complexes in yeasts, with major differences with the same complexes in mammals and flies. The evidence supporting the conclusions is **convincing**, with rigorous high-throughput sequencing approaches and detailed functional analysis. This work will be of broad interest to colleagues in the RNA modification and meiosis fields.

## Introduction

The *N6*-methyladenosine (m6A) modification is the most widespread internal RNA modification on mammalian messenger RNA (mRNAs). The mark is deposited by a multi-subunit protein complex, also known as the m6A *m*ethyl*t*ransferase ('writer') *c*omplex (MTC), of which the catalytic subunit is METTL3 in human and Ime4 in yeast (*Balacco and Soller, 2019*; *Clancy et al., 2002*). In mammals, the deposition of m6A occurs at DRACH motifs throughout transcripts that are distant from splice sites,

**Figure 1.** Identification of Mum2-interacting proteins. (**A**). Scheme of methyltransferase complexes (MTCs) in mammals, *Drosophila*, *Arabidopsis,* and yeast. Colour indicates matching orthologues. (**B**) Volcano plots of Mum2-TEV-ProA compared to untagged control. Diploid cells harbouring Mum2 tagged with TEV-ProA (FW7873) or untagged control (FW1511) were induced to enter meiosis. Protein extracts were incubated with ProA-coated paramagnetic beads. TEV protease was used to elute Mum2 from the beads. Significantly enriched proteins are labelled in blue and known subunits of the MIS complex are labelled in red. (**C**) Similar analysis as in (**B**), except that during IP extracts were either untreated (left panel) or RNase treated (right panel). (**D**) m6A levels as determined by LC-MS in WT diploid cells, or diploid cells harbouring *ime4Δ*, *mum2Δ*, or *slz1Δ* (FW1511, FW7030, FW6535, and FW6504). In short cells were grown to saturation in rich medium (YPD), shifted to BYTA and grown for another 16 hr, then shifted to sporulation medium (SPO). Samples were taken at 4 hr in SPO. Total RNA was extracted and followed by two rounds of polyA RNA purification. Subsequently, RNA was digested into nucleosides and m6A was quantified my LC-MS. The means of n = 3 biological replicates are shown. (**E**) Same analysis as in (**D**), except that multiple time points after shifting cells to SPO were taken (0, 2, 4, 6, 8, 10, 12 hr in SPO). WT, *slz1Δ*, or *ime4Δ* diploid cells were used. The means of n = 2 biological repeats are shown. (**F**) m6A levels in WT, *ime4Δ*, and *kar4Δ*cells (FW1511, FW7030, FW8246). Samples were collected at 4 hr in SPO. The means of n = 3 biological replicates are shown.

The online version of this article includes the following figure supplement(s) for figure 1:

**Figure supplement 1.** Identification of Mum2-interacting proteins.

biasing this modification towards long internal and last exons (***Dominissini et al., 2012***; ***Meyer et al., 2012***; ***Uzonyi et al., 2023***). The presence of m6A on mRNA primarily impacts mRNA stability but has also been linked to a wide array of additional molecular outcomes, and its disruption is associated with diverse phenotypes ranging from immune dysfunction to cancer (***Jiang et al., 2021***; ***Murakami and Jaffrey, 2022***; ***Zaccara et al., 2019***).

METTL3, which forms a hetero dimer complex with METTL14, is the catalytic unit of the MTC (***Liu et al., 2014***; ***Wang et al., 2016***). Several auxiliary proteins that interact with METTL3/METTL14 are required for the deposition of m6A. These include the scaffold proteins WTAP and VIRMA, E3 ligase HAKAI, ZCH13H3, and RNA binding proteins RBM15/RBM15b (***Bawankar et al., 2021***; ***Knuckles et al., 2018***; ***Ping et al., 2014***; ***Schwartz et al., 2014***; ***Wang et al., 2021***; ***Wen et al., 2018***; ***Yue et al., 2018***). Structural and topological analyses of the mammalian and *Drosophila* MTCs revealed that MTCs can be divided into two subcomplexes: (1) the m6A METTL complex (MAC) which consists

of METTL3/METTL14, and an (2) m6A-METTL-associated complex (MACOM) consisting of WTAP, VIRMA, HAKAI, ZCH13H3, and RBM15/RBM15b (*Figure 1A*; *Lence et al., 2019*; *Su et al., 2022*). MACOM directly interacts with MAC to enhance m6A deposition. In *Arabidopsis thaliana,* all subunits of both MAC and MACOM, except for RBM15/RBM15b, are conserved and play related roles in m6A deposition, suggesting structural and topological conservation in plants (*Růžička et al., 2017*; *Zhang et al., 2022*).

In yeast, m6A is deposited solely during early meiosis (*Agarwala et al., 2012*; *Shah and Clancy, 1992*). Intriguingly, early dissections of the yeast MTC have revealed only three proteins *Mum2, Ime4,* and *Slz1,* collectively coined the MIS complex (*Agarwala et al., 2012*; *Figure 1A*). However, only two of these proteins (Ime4 and Mum2) have mammalian homologues (METTL3 and WTAP respectively), which could suggest that MTCs have undergone substantial rewiring throughout evolution.

More than 1000 mRNAs are methylated during early yeast meiosis (*Schwartz et al., 2013*). Like in mammals, m6A-modified mRNAs are turned over more rapidly compared to unmodified mRNAs (*Bushkin et al., 2019*; *Scutenaire et al., 2022*; *Varier et al., 2022*). The decay of m6A transcripts requires active translation and is mediated by the YTH domain containing protein Pho92/Mrb1 (*Varier et al., 2022*). Several studies have shown that preventing the m6A modification at single meiotic transcripts can cause defects in meiosis (*Bushkin et al., 2019*; *Scutenaire et al., 2022*). However, the exact function of the m6A modification in yeast meiosis remains elusive. In yeast, Ime4 has both m6A-dependent and m6A-independent functions in meiosis as a catalytically inactive mutant of Ime4 shows milder delay in meiotic progression compared to the Ime4 deletion mutant (*Agarwala et al., 2012*; *Clancy et al., 2002*).

The fact that MTCs in mammals, *Drosophila,* and plants harbour at least six conserved protein subunits (*Figure 1A*) motivated us to re-examine the composition of the yeast MTC. Here we uncover and characterize three novel components of the yeast MTC (Kar4, Ygl036w/Vir1, and Dyn2). We further demonstrate that Kar4 and Vir1, and potentially also Slz1, a previously known component, are orthologues of known components of the mammalian MTC (METTL14, VIRMA, and ZCH13H3, respectively). In total, five members of the yeast MTC likely have orthologues matching the mammalian MTC. We show that Ime4, Mum2, Vir1, and Kar4 form a stable complex on mRNAs required for meiotic progression, while the Slz1 subunit is essential for m6A deposition but not for MTC integrity. The composition of the yeast MTC is considerably more similar to its mammalian counterparts than previously thought. Our findings also suggest that in contrast to mammals and *Drosophila*, the yeast MTC has no MAC and MACOM arrangement.

## Results
### Identification of Mum2 (and Kar4)-interacting proteins

The inconsistency in the composition of the yeast MTC, comprising only three known components (*Agarwala et al., 2012*), in comparison to its mammalian, *Drosophila,* and plant MTC counterparts all comprising at least six conserved proteins (seven in the case of mammals and *Drosophila*), prompted us to re-evaluate the yeast MTC using a proteomics approach (*Figure 1A*). We generated a yeast strain with the conserved scaffold protein in MTCs, Mum2, fused to the ProA-tag followed by a TEV cleavage sequence at the carboxy terminus. Subsequently, we performed immunoprecipitation with the ProA-tag in protein extracts generated from cells staged in early meiosis followed by elution with the TEV protease (*Figure 1B* and *Supplementary file 1*). As expected, Ime4 was strongly enriched in the Mum2 IP-MS analysis. We also identified two other proteins that were strongly enriched, Ygl036w and Kar4. Two additional RNA-binding proteins, Pab1 and Npl3, and light chain dynein protein, Dyn2, were significantly enriched in the Mum2 IP, albeit to a lesser extent ($\log_2$ protein intensity fold change [FC] > 2, and -log *t*-test p-value>1). Noteworthy, Slz1, a known component of the MIS complex, was not significantly enriched (*Figure 1B*).

To confirm that Kar4 and Ygl036w are integral parts of the yeast MTC, we performed a Kar4 IP-MS using a similar setup (*Figure 1—figure supplement 1A*). Even though Kar4 enrichment was only four-fold over background, we found that Mum2, Ime4, and Ygl036w, but not Slz1, significantly co-purified with Kar4 ($\log_2$ protein intensity FC > 2, and -log *t*-test p-value>1).

Next, we examined whether the interaction with Mum2 was dependent on the presence of RNA in the sample by treating the protein lysate with RNase. The Mum2-IP MS analysis identified Ime4,

Ygl036w, Kar4, Slz1, and Dyn2 in both control and RNAse-treated samples (*Figure 1C*, *Figure 1—figure supplement 1B*, and *Supplementary file 1*). Pab1 was identified in the control but was not enriched in the RNAse-treated sample, while Npl3 was not enriched at all in these analyses (log$_2$ protein intensity FC > 2, and -log *t*-test p-value>1) (*Figure 1C*, *Figure 1—figure supplement 1B*, and *Supplementary file 1*). We conclude that Ime4, Ygl036w, Kar4, Slz1, and Dyn2 interact with Mum2 in an RNA-independent manner, while Pab1 interaction with Mum2 is RNA-dependent and hence indirect. These data suggest that Kar4, Ygl036w, and Dyn2 are MTC components.

## Kar4, Ygl036w, and Slz1 are essential for m6A deposition

Next, we determined whether known members and newly identified interactors are required for m6A deposition. Previous work showed that m6A deposition is severely reduced in deletion mutants of two components of the MIS complex: *IME4* and *MUM2* (*Agarwala et al., 2012*). To confirm these previous findings, we determined m6A levels in diploid cells undergoing early meiosis (4 hr in sporulation medium [SPO]) harbouring gene deletion in *MUM2* and *IME4* (*mum2Δ* and *ime4Δ*). Both deletion mutants showed a severe reduction in m6A over A levels as determined by LC-MS (*Figure 1D*), consistent with previous measurements (*Ensinck et al., 2023*). We also quantified m6A levels in an *SLZ1* deletion strain. We found that *slz1Δ* cells showed a similar reduction in m6A levels as *ime4Δ* cells (LC-MS), suggesting that Slz1 is also required for m6A deposition (*Figure 1D*). The analysis contrasts previous work that showed only a partial reduction in m6A levels in *slz1Δ* cells (*Agarwala et al., 2012*). One possibility for the discrepancy is that *slz1Δ* cells had a delay in m6A accumulation during early meiosis, and possibly m6A accumulation occurs later in meiosis in *slz1Δ* cells. However, m6A levels in *slz1Δ* cells did not accumulate at any time in the 12 hr following meiotic induction and showed background levels similar to *ime4Δ* cells (*Figure 1E*).

In yeast, Kar4 is known to act as a transcription factor important for the mating response pathway; however, no clear mechanistic role for Kar4 in meiosis has been reported (*Kurihara et al., 1996*). Interestingly, high-throughput studies suggest that the *KAR4* deletion negatively affects meiosis and sporulation (*Deutschbauer et al., 2002*; *Enyenihi and Saunders, 2003*). Based on sequence homology analysis, Kar4 has been proposed to be the yeast orthologue of METTL14, a critical component of MTCs in mammals, *Drosophila*, and plants. Phylogenetic analysis of methyltransferase domains showed that Kar4 and METTL14 belong to the same subfamily, but are relatively different from each other (*Bujnicki et al., 2002*). Since METTL14 is considered a structural subunit in the METTL3/METTL14 heterodimer, we hypothesized that Kar4 may play a related role in m6A deposition (*Śledź and Jinek, 2016*; *Wang et al., 2016*). We found that like *ime4Δ* cells entering meiosis, *kar4Δ* cells showed no detectable m6A levels (*Figure 1F*).

To examine whether Ygl036w, Dyn2, and the other identified Mum2-interacting proteins (Pab1 and Npl3) are important for m6A deposition, we performed m6A-ELISA and m6A-seq2 (*Figure 2*). Both analyses by m6A-ELISA and m6A-seq2 showed that *ygl036wΔ* had no detectable m6A levels, similar to *ime4Δ* (*Figure 2A and B*). Additionally, *dyn2Δ* had reduced m6A levels (50% of WT) as detected by m6A ELISA (*Figure 2C*). Pab1 is an essential gene and Npl3 is essential for entry into meiosis. To examine whether the two are required for m6A deposition, we depleted both proteins using the auxin-induced degron (AID) system during meiosis (Figure S2). To obtain efficient depletion in meiosis, we let cells enter early meiosis (2 hr SPO) and subsequently induced depletion of Pab1 (*PAB1-AID*) or Npl3 (*NPL3-AID*) for 2 hr (*Figure 2—figure supplement 1*). As a control, we also depleted Slz1 (*SLZ1-AID*). We found that depletion of Pab1 and Npl3 showed no significant reduction in m6A levels compared to the control (*Figure 2D*). As expected, depletion of Slz1 showed a strong reduction in m6A. Also given that Pab1 and Npl3 interactions with Mum2 were RNA-dependent or not reproducible (in the case of Npl3), both RNA-binding proteins are likely not part of the yeast MTC. We conclude that in addition to Ime4 and Mum2, Slz1, Kar4, and Ygl036w are all essential for m6A deposition.

## The m6A writer complex is conserved

Our analysis revealed that the yeast MTC comprises at least six protein subunits (Mum2, Ime4, Kar4, Ygl036w, Slz1, Dyn2), all of which – with the exception of Dyn2 – are required for m6A methylation. Two subunits that were described previously have mammalian orthologues: Ime4/METTL3 and Mum2/WTAP. Among the newly identified subunits, Kar4 has a clear mammalian orthologue, named METTL14.

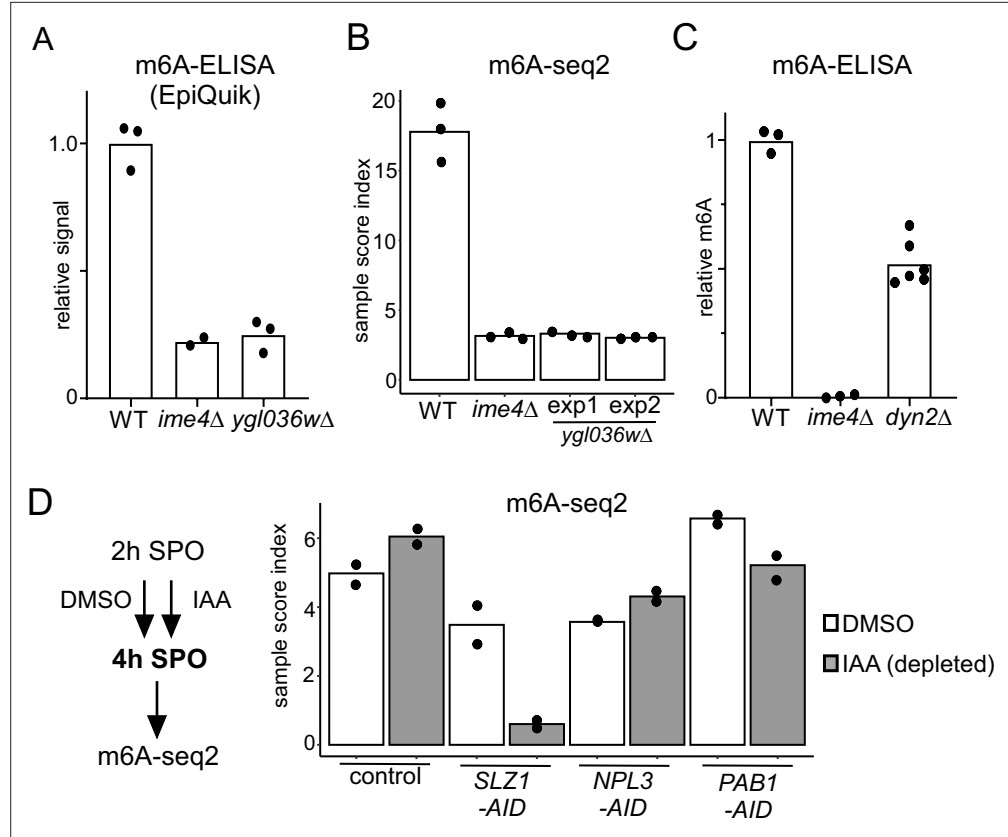

**Figure 2.** Functional characterization of Mum2 interactors. (**A**) WT, *ime4Δ*, and *ygl036wΔ* cells (FW1511, FW7030, FW9307) were induced to enter meiosis. RNA was extracted, and polyA RNA was purified. m6A levels were determined by m6A-ELISA kit supplied by Epiquik. The means of n = 3 biological replicates are shown. (**B**) m6A-seq2 analysis of strains and condition described in (**A**). Two independent clones of *ygl036wΔ* were used for the analysis. Shown are the sample index score representing 1308 known of m6A sites. The means of n = 3 biological repeats are shown. (**C**) m6A levels of WT, *ime4Δ*, and *dyn2Δ* cells (FW1511, FW7030, FW10442). RNA was extracted, and polyA RNA was purified. m6A levels were determined by a non-commercial m6A-ELISA assay (***Ensinck et al., 2023***). The means of n = 3 biological replicates are shown for WT and *ime4Δ*, and n = 6 for *dyn2Δ*. (**D**) m6A-seq2 analysis after Npl3, Pab1, and Slz1 depletion. Diploid cells containing *PAB1-AID*, *NPL3-AID*, *SLZ1-AID* (FW10388, FW10389, FW10386) were treated at 2 hr in SPO for 2 hr with DMSO or IAA and CuSO4 for rapid depletion. A control strain was included that only harboured the TIR1 ligase expressed from the *CUP1* promoter (FW5737). Shown are the m6A-sample indices, quantifying the overall levels of enrichment over 1308 previously defined m6A sites. The means of n = 2 biological repeats are shown.

The online version of this article includes the following source data and figure supplement(s) for figure 2:

**Figure supplement 1.** Functional characterization of Mum2 interactors.

**Figure supplement 1—source data 1.** Western blot of auxin-induced degron (AID) depletion.

**Figure supplement 1—source data 2.** Western blot of loading control Hxk1.

**Figure supplement 1—source data 3.** LI-COR scan auxin-induced degron (AID) and Hxk1.

The two remaining subunits, Ygl036w and Slz1, did not have any obvious homologues based on standard BLAST searches. To investigate this further, we first conducted pairwise comparisons of the Ylg036w protein coding sequence against all sequences of the human proteome, relying on both global (Needle) and local (Matcher) sequence similarity scores (***Figure 3A***). The analysis revealed that Ygl036w displayed a striking sequence similarity to VIRMA, a known subunit of the mammalian MTC. We next performed proteome-wide structural alignments of the predicted structure of Ygl036w against all predicted structures of the human proteome. The top hit in this analysis, by far, was VIRMA (***Figure 3B***). A structural alignment of Ygl036w against VIRMA displays a high extent of agreement (TM-score of 0.483, normalized for Ygl036w) (***Figure 3C***, ***Figure 3—figure supplement 1***). These

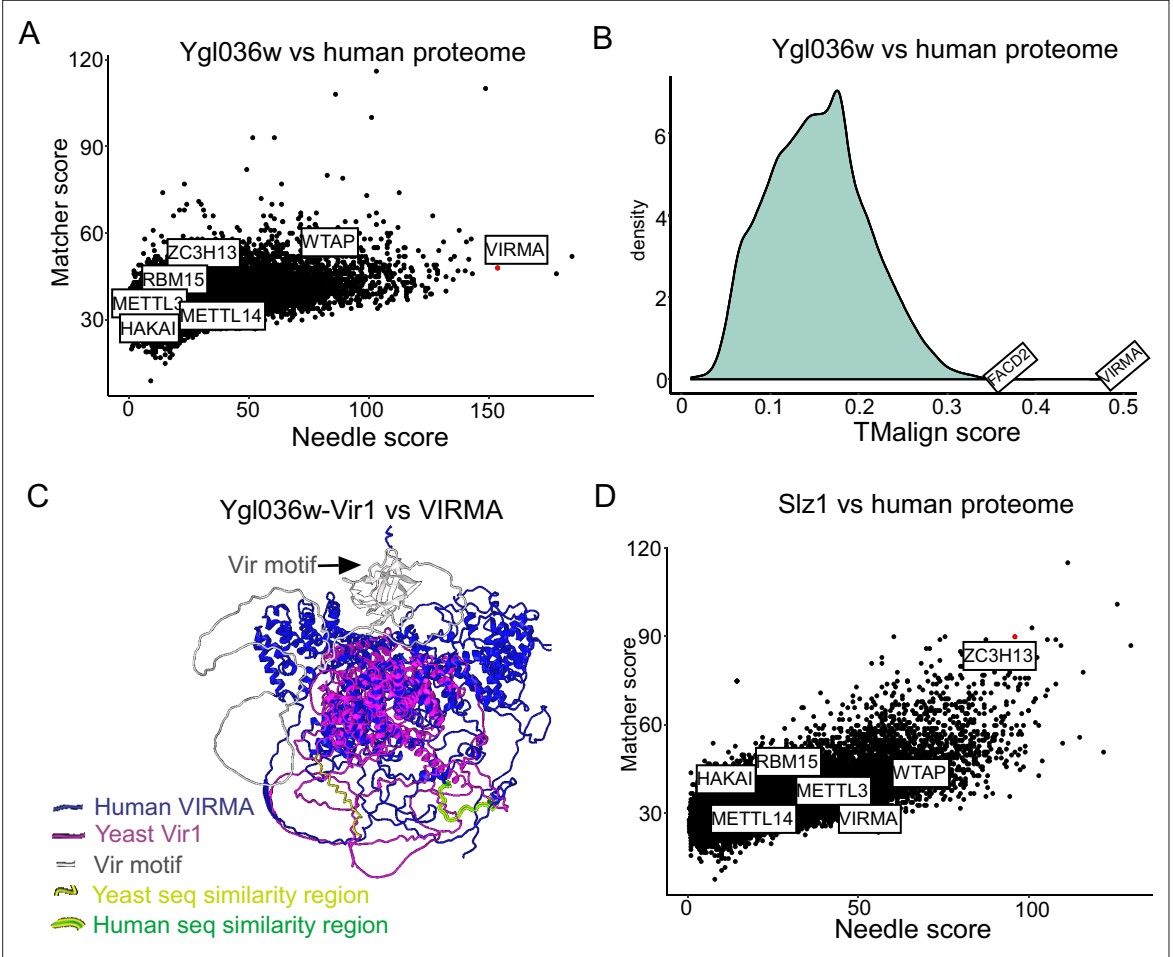

**Figure 3.** Ylg036w-Vir1 and Slz1 have orthologues in humans. (**A**) Pairwise sequence comparison of Ygl036w against all human protein coding genes. Global similarity scores (using Needle) are displayed on the x-axis, whereas local similarity scores (using Matcher) are displayed on the y-axis. (**B**) Pairwise structural similarity of Ygl036w against all human protein coding genes, using TMalign algorithm. A histogram of the TMalign density scores is displayed, along with an indication of the score against VIRMA, the top hit in this analysis, and the second highest hit, FACD2. TM-scores were calculated and normalized using the query gene length, Ygl036w. (**C**) Overlay of Alpha-fold predicted structures of Ygl036c-Vir1 and human VIRMA. Human VIRMA is shown in blue and Yeast Ygl036c-Vir1 is shown in pink. Also indicated are the Vir motif and regions with strong sequence similarities. (**D**) Analysis as in (**A**) using Slz1 as a query.

The online version of this article includes the following figure supplement(s) for figure 3:

**Figure supplement 1.** Ylg036w-Vir1 and Slz1 have orthologues in humans.

analyses thus strongly suggest that Ygl036w and VIRMA are homologues, and we hence renamed Ygl036W to Vir1, as the yeast orthologue of VIRMA in mammals, Virilizer in *Drosophila*, and VIR in plants, respectively.

Motivated by these results, we next sought to conduct the same sets of analyses for Slz1. Strikingly, Slz1 showed partial similarity with ZC3H13 (*Figure 3D*). ZC3H13 is highly unstructured, and much larger than Slz1, which made 3D structural alignments uninformative. Interestingly, both SLZ1 and ZC3H13 have been proposed to play a similar role, of shuttling the MTC complex into the nucleus (*Schwartz et al., 2013*; *Wen et al., 2018*), providing further evidence that these two proteins may be orthologues. We conclude that the yeast MTC is considerably more conserved than previously described, with five subunits of the mammalian MTC complex (METTL3, METTL14, WTAP, VIRMA, ZC3H13) likely having orthologues in the yeast counterpart (Ime4, Kar4, Mum2, Vir1, Slz1).

## Topological features of the yeast MTC

To gain further insight on the organization of the yeast MTC, we dissected the interrelationships between the MTC components. Often, protein subunits interacting as stable macromolecular

complexes are destabilized when protein complexes are disrupted, for example, upon deletion of a scaffold protein. With this in mind, we determined protein expression levels of all yeast MTC subunits in deletion mutants for each of the remaining subunits (*Figure 4A*, *Figure 4—figure supplement 1A–E*). The analyses revealed that Ime4, Mum2, and Vir1 were all required for mutual stabilisation in cells undergoing meiosis, with protein levels of any of these three components being strongly reduced upon elimination of any of the other two (*Figure 4A and B*, *Figure 4—figure supplement 1A–C*). Interestingly, loss of any of these proteins also led to loss of Kar4, yet – surprisingly – this was not mutual, as *kar4Δ* had little effect on the expression of any of the subunits (*Figure 4A and B*, *Figure 4—figure supplement 1D*). The *slz1Δ* had little effect on the expression of any of the subunits (*Figure 4A and B*, *Figure 4—figure supplement 1A–D*). Importantly, loss of the different components was not due to changes in RNA levels as shown by RNA-seq as these remain relatively unchanged in the mutants except for Slz1, where *SLZ1* mRNA induction was reduced in the deletion mutants resulting in lower Slz1 protein levels (*Figure 4A and B*, *Figure 4—figure supplement 1E and F*). These results are at odds with results in mammalian cells where METTL14, the Kar4 orthologue, stabilizes METTL3, and VIRMA is not important for METTL3 and METTL14 stabilization (*Wang et al., 2016*; *Wang et al., 2014*; *Yue et al., 2018*).

To determine the manifestation of these interdependencies in the composition of the m6A writer complex, we performed Mum2 IP-MS in WT, *slz1Δ*, and *kar4Δ* cells (*Figure 4C* and *Supplementary file 1*). In line with the protein expression analysis of the m6A writer complex deletion mutants, we found that in *slz1Δ* cells the Mum2-Ime4-Vir1-Kar4 complex remained intact, and in *kar4Δ* cells Mum2 formed interactions with Vir1 and Ime4. Slz1 and Dyn2 were also both enriched in the Mum2-IP in *kar4Δ* cells, suggesting that Kar4 does not mediate interactions between the other subunits of the m6A writer complex (*Figure 4C* and *Supplementary file 1*). Interestingly, Dyn2 was not identified as an interactor with Mum2 in the *slz1Δ* cells, suggesting that Slz1 mediates the interaction of Dyn2 with the MTC (*Figure 4C* and *Supplementary file 1*).

Our analyses showed that in cells undergoing meiosis Ime4, Mum2, Vir1, Kar4, Slz1, and Dyn2 make up the MTC, and that, except for Dyn2, all MTC components are essential for m6A deposition. However, it does not exclude the possibility that other proteins might be required for MTC assembly. Therefore, we co-expressed all five subunits essential for m6A (Ime4, Kar4, Mum2, Vir1, Slz1) in insect cells. Subsequently, we performed a single-step affinity purification using the Strep-tag system with a Strep-tag II fused to the amino-terminal domain of Ime4 (*Figure 4D*). The analyses revealed that Vir1, Mum2, Kar4, and Slz1 co-purified with Ime4 (*Figure 4E*), showing that these five subunits form a stable complex in vitro.

Collectively, these analyses suggest that the yeast MTC comprises three mutually stabilizing components: (1) a core comprising Ime4, Mum2, and Vir1; (2) Kar4, which also is stabilized when part of the MTC; and (3) a module comprising Slz1 and Dyn2, where Slz1 acts as a hub for binding of Dyn2. Additionally, we showed that the MTC components, Ime4, Mum2, Vir1, Kar4, and Slz1, form a stable complex in vitro.

## The yeast MTC has m6A-dependent and -independent functions in meiosis

It has previously been established that Ime4, the catalytic component, has both m6A-dependent and m6A-independent functions on the basis of the observation that an Ime4 catalytic inactive mutant (*IME4*^CD) displays a milder phenotype than an *IME4* deletion mutant (*Agarwala et al., 2012*; *Clancy et al., 2002*). We confirmed the difference between *ime4Δ* and *IME4*^CD. Indeed, we found that *IME4*^CD showed no m6A deposition and a milder delay in meiosis compared to the *ime4Δ* (*Figure 5—figure supplement 1A and B*). One can conceive of two classes of m6A-independent roles for Ime4: (1) a 'moonlighting' function, independent also of the role of Ime4 in forming part of the yeast MTC; and (2) an MTC-dependent (yet m6A-independent) function. We reasoned that the availability of three (Vir1, Kar4, Dyn2) new additional components of the MTC complex might allow us to distinguish between these possibilities. If the m6A-independent role of Ime4 is independent of the MTC, deletion of *IME4* would result in a phenotype unique to Ime4 and not overlapping with its counterparts upon deletion of other MTC components. In contrast, if the phenotype is due to the requirement of the MTC complex, identical phenotypes should be observed also upon loss of additional components.

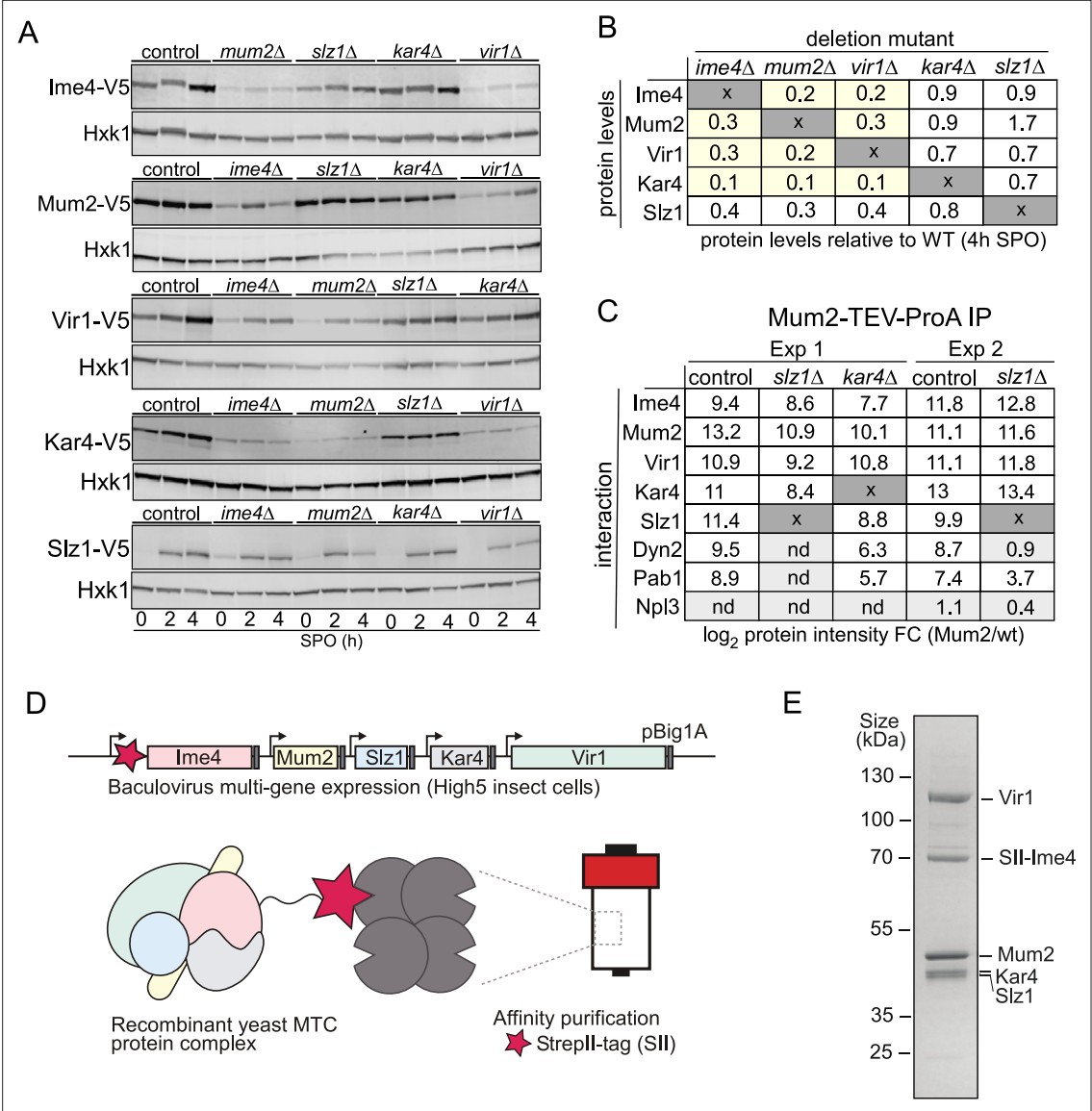

**Figure 4.** Topological and structural properties of the yeast methyltransferase complexes (MTC). (**A**) Ime1, Mum2, Vir1, Kar4, and Slz1 expression depends on the presence of MTC components. WT, *mum2*D, *ime4Δ*, *slz1Δ*, *kar4Δ*, or *vir1Δ* were examined for Ime4-V5 (first panel, FW6057, FW8362, FW8264, FW8633, and FW9483), Mum2-V5 (second panel, FW6500, FW9394, FW6534, FW9396, and FW9398), Vir1-V5 (third panel, FW9666, FW9663, FW9668, FW9670, and FW9643), Kar4-V5 (fourth panel, FW8216, FW9484, FW8212, FW9482, and FW9481), or Slz1-V5 (fifth panel, FW6502, FW9479, FW9477, FW9480, and FW9478) during early meiosis. Samples were taken at the indicated time points. Western blots were probed with anti-V5 antibodies and anti-Hxk1 as a loading control. (**B**) Table of western blot quantifications of the 4 hr time point described in (**C**). Each row represents the protein expression (Ime4-V5, Mum2-V5, Vir1-V5, Kar4-V5, and Slz1-V5) and each column the deletion mutant (*mum2*D, *ime4Δ*, *slz1Δ*, *kar4Δ*, or *vir1Δ*). Highlighted in yellow are the MTC components affected in protein levels (<0.5), but not at the mRNA levels. (**C**) Diploid cells untagged or harbouring Mum2 tagged with TEV-ProA in WT, *slz1Δ*, or *kar4Δ* background (FW1511, FW7873, FW10158, and FW10159) were induced to enter meiosis. Protein extracts were incubated with ProA-coated paramagnetic beads. TEV protease was used to elute Mum2 from the beads. Shown are the enrichments for Ime4, Mum2, Vir1, Kar4, Slz1, Pab1, and Npl3. Two independent experiments are shown for *slz1Δ*. nd = not detected. (**D**) Purification of the five-subunit recombinant m6A writer complex. Schematic representation showing the baculovirus recombinant co-expression and purification of the yeast MTC with five subunits (Ime4, Mum2, Kar4, and Slz1) in insect cells. Genes are depicted by rectangles, promoters by arrows, and terminators by grey boxes. The star in magenta represents the Strep-tag in the N-terminal domain of Ime4. (**E**) SDS-PAGE analysis of the purified recombinant m6A writer complex after affinity. Identities of bands confirmed by MS are labelled.

The online version of this article includes the following source data and figure supplement(s) for figure 4:

**Source data 1.** LI-COR scan Ime4-V5 and Hxk1.

**Source data 2.** LI-COR scan Mum2-V5 and Hxk1.

*Figure 4 continued on next page*

*Figure 4 continued*

**Source data 3.** LI-COR scan Vir1-V5 and Hxk1.

**Source data 4.** LI-COR scan Kar4-V5 and Hxk1.

**Source data 5.** LI-COR scan Slz1-V5 and Hxk1.

**Source data 6.** Uncropped gel.

**Figure supplement 1.** Topological and structural properties of the yeast methyltransferase complexes (MTC).

Accordingly, we monitored the progression of meiosis upon loss of each of the MTC components. *kar4Δ*, *vir1Δ*, *mum2Δ*, and *ime4Δ* cells all displayed a strong defect in meiosis, and <40% completed meiotic divisions after 24 hr in SPO (*Figure 5A*). In contrast, *slz1Δ* had a considerably milder phenotype in meiosis compared to *ime4Δ* cells (*Figure 5A*; *Agarwala et al., 2012*). *dyn2Δ* showed the mildest phenotype in meiosis with only marginal delay in meiotic divisions and little effect on the number of cells that underwent meiosis after 24 hr in SPO (*Figure 5B*).

To characterize the role played by the different writer complex subunits during meiosis in greater detail, we profiled RNA expression in each of the MTC mutants along a high-resolution meiotic time course. We took samples at 0, 2, 4, 6, and 8 hr following meiotic induction, which covers the early and middle phases of the yeast sporulation programme. For the analysis, we compared WT, *ime4Δ*, *mum2Δ*, *slz1Δ*, *kar4Δ*, *vir1Δ*, and *dyn2Δ*. The experiment allowed us to assess more precisely which stage of meiosis was affected in the different deletion mutants of the m6A writer complex (*Supplementary file 2*). Two approaches were taken to analyse the data. First, we used a principal component analysis (PCA) and hierarchical clustering across all time points and all mutants (*Figure 5C*, *Figure 5—figure supplement 1C*). Second, we assessed gene expression across different gene clusters in meiosis described previously (*Chu et al., 1998*). These genes clusters represent the early metabolic genes (cluster C1), early meiotic genes (clusters C2 and C3), early to middle meiotic genes (cluster C4), and middle meiotic genes (cluster C5), respectively (*Figure 5D*). Consistent with the meiotic phenotypes, we found that *kar4Δ*, *vir1Δ*, *mum2Δ*, and *ime4Δ* displayed the most severe expression phenotypes (*Figure 5C and D*, *Figure 5—figure supplement 1C*). This is most pronounced between the 4-to-6-hr time points, where *kar4Δ*, *vir1Δ*, *mum2Δ*, and *ime4Δ* remained staggered at cluster C2, whereas WT transitioned into clusters C4 and C5. The fact that the different MTC mutants clustered together is highly suggestive of a function common to all of them, as opposed to moonlighting functions of individual components (*Figure 5C and D*, *Figure 5—figure supplement 1C*). Also consistent with the phenotypic analysis, *slz1Δ* showed a mild delay, distinct from the remaining mutants, whereas *dyn2Δ* showed little difference in comparison to WT (*Figure 5C and D*, *Figure 5—figure supplement 1C*).

Interestingly, despite the general lack of differences between the strains at the 2 hr time point, the histone genes, *HHF2* and *HTB2*, were significantly reduced in expression in five of the deletion mutants (*ime4Δ*, *mum2Δ*, *slz1Δ*, *kar4Δ*, *vir1Δ*), potentially hinting at an early role of the MTC complex in regulation of histone levels, reminiscent of recent findings in planaria (*Figure 5—figure supplement 1D*; *Dagan et al., 2022*).

Finally, we determined whether the m6A-dependent decay function is reflected in the RNA-seq dataset. Recent work showed that the sole m6A reader protein in yeast Mrb1/Pho92 controls the decay of m6A-modified mRNAs during meiosis (*Varier et al., 2022*). We used an iCLIP dataset that identified the mRNAs associated by Pho92 in an m6A-dependent manner to assess how these mRNAs were affected throughout the time course in WT and deletion mutants (*Varier et al., 2022*). Using this subset of the transcripts, we observed that in WT cells these transcripts were generally induced up to the 2 hr time point, following which their levels declined. In contrast, in all five deletion mutants essential for m6A (*ime4Δ*, *mum2Δ*, *slz1Δ*, *kar4Δ*, *vir1Δ*), their levels failed to decline after their initial induction at 2 hr (*Figure 5E*). Consistent with the m6A and meiotic phenotype, the *dyn2Δ* only showed marginal difference compared to the WT (*Figure 5E*). A control set of randomly chosen transcripts did not display a difference between the WT and the deletion mutants (*Figure 5—figure supplement 1E*). These data support the m6A-dependent function of the MTC complex subunits, which is to m6A modify mRNAs for decay via Pho92 during early yeast meiosis (*Varier et al., 2022*).

Taken together, these analyses suggest that Kar4, Mum2, Vir1, and Ime4 play a dual role in controlling the expression program of meiosis via both m6A-dependent and m6A-independent yet MTC-complex-dependent functions, in contrast to Slz1 exhibiting only an m6A-dependent function.

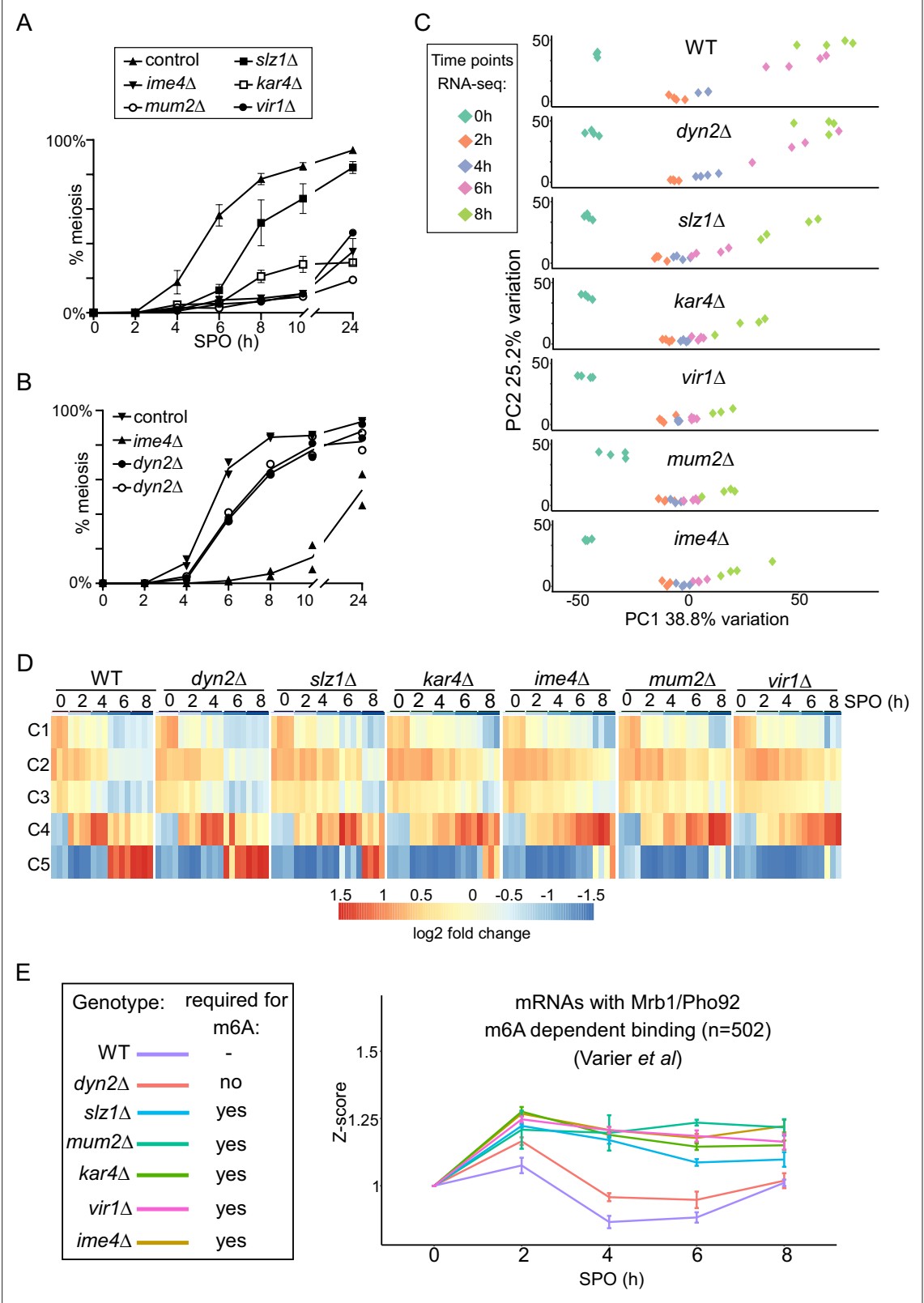

**Figure 5.** *N6*-methyladenosine (m6A)-dependent and -independent roles of the methyltransferase complexes (MTC) in meiosis. (**A**) Onset of meiosis in WT, *ime4Δ*, *mum2Δ*, *slz1Δ*, *kar4Δ*, and *vir1Δ* (FW1511, FW7030, FW6535, FW6504, FW8246, and FW9307). Cells induced to enter meiosis. Samples were taken at the indicated time points for DAPI staining. Cells were fixed, stained with DAPI, and nuclei were counted for at least 200 cells per biological repeat. Cells with two or more DAPI masses were considered to undergo meiosis. The mean and SEM of n = 3 biological repeats are

*Figure 5 continued on next page*

*Figure 5 continued*

displayed. (**B**) Similar analysis as (**A**), except that the WT, *ime4Δ*, and *dyn2Δ* were analysed (FW1511, FW7030, FW10442, and FW10443). For the analysis, two independent *dyn2Δ* strains were used. The means of n = 2 biological repeats are displayed. (**C**) Principal component analysis (PCA) of gene expression measurements across diverse strains and time points, done using PCAtools 2.6.0 for R. Strains described in (**A**) and (**B**) were induced to enter synchronized meiosis and samples were taken at the indicated time points (0, 2, 4, 6, and 8 hr in sporulation medium [SPO]). For analysis, n = 4 biological repeats were used, except for *ime4Δ*_4h, *ime4Δ*_0h, and *vir1Δ*_8h, which had an n = 3. (**D**) Median expression patterns in WT and MTC mutants of five predefined gene clusters representing early metabolic genes (C1, n = 67), early meiotic genes (C2, n = 42; C3, n = 34), early to middle meiotic genes (C4, n = 53), and middle meiotic genes (C5, n = 136) (*Chu et al., 1998*). (**E**) Median gene expression fold change from time point 0 of Mrb1/Pho92-bound transcripts during the time course across the different deletion mutants. Dataset of Mrb1/Pho92 iCLIP-based targets associating with RNA in an m6A-dependent manner identified in *Varier et al., 2022* was used for the analysis. For each transcript, a Z-score was calculated with respected to the 0 hr time point. The means of at least n = 3 biological replicates are shown, and error bars represent the standard error of the mean.

The online version of this article includes the following figure supplement(s) for figure 5:

**Figure supplement 1.** *N6*-methyladenosine (m6A)-dependent and -independent functions of the methyltransferase complexes (MTC) in meiosis.

## The molecular functions of Kar4 in meiosis and mating differ

Kar4 has been shown to be important for controlling transcription of genes involved in the mating pathway in yeast. When haploid cells sense mating pheromone, Kar4 is induced and is recruited to promoters predominantly controlled by the transcription factor Ste12 to activate transcription of genes involved in the mating pathway (*Gammie et al., 1999*; *Kurihara et al., 1996*; *Lahav et al., 2007*). Our above-described findings reveal that Kar4 is also an integral component of the yeast MTC and important for meiosis. We therefore wondered whether the mating and meiotic functions might be mediated via a shared mechanism. Under such a scenario, Kar4's roles in mating might require the additional MTC components, and – conversely – the role of Kar4 in meiosis might be mediated via binding to chromatin. Under the latter scenario, we speculated that Kar4 might function in directing the MTC to chromatin to facilitate co-transcriptional m6A deposition, which could be consistent with one of the proposed modes of action of METTL14 in mammals (*Huang et al., 2019*).

To examine this, we first tested whether Ime4, Slz1, or Mum2 are expressed during mating. As expected, a shorter protein isoform of Kar4 was strongly induced when *MATa* cells were treated with mating pheromone (α-factor) (*Figure 6A*; *Gammie et al., 1999*). However, Ime4 and Mum2 expression was not elevated and Slz1 expression was not detected at all upon pheromone treatment. We also examined whether Kar4 expression during mating was dependent on the components of the MIS complex as observed in meiosis. We found that Kar4 levels were unaffected in *ime4Δ*, and *mum2Δ* cells (*Figure 6B*). In line with protein expression data, we found that Kar4 was bound to the *AGA1* promoter upon mating pheromone treatment as determined by ChIP, while other MTC subunits (Ime4, Mum2, Slz1) showed no enrichment (*Figure 6C*; *Aymoz et al., 2018*). These data indicate that the yeast MTC does not play a role in the mating pathway.

Second, we determined Kar4 localization during meiosis and mating by using a strain with Kar4 fused to mNeongreen (Kar4-mNG). We found that the Kar4 localization pattern was different between mating and meiosis. Kar4-mNG concentrated inside the nucleus in *MATa* cells treated with α-factor (*Figure 6D and E*, *Figure 6—figure supplement 1A*). During meiosis (*MATa/α* cells), however, we found that Kar4-mNG was localized to both the cytoplasm and the nucleus (*Figure 6D and E*, *Figure 6—figure supplement 1B*).

Third, we examined whether Kar4 or several components of the yeast MTC (Ime4, Mum2, and Szl1) associate with chromatin during meiosis (*Figure 6F and G*, *Figure 6—figure supplement 1C and D*). While Kar4 was bound to 800 genomic locations in *MATa* cells treated with α-factor, only a few Kar4 peaks were detected during meiosis in *MATa/α* cells (<10) (*Figure 6F and G*). As expected, Kar4 ChIP-seq peaks were enriched for the Ste12-binding sites (motif 3 in *Figure 6H*), but other motifs were also identified, perhaps indicating that Kar4 can bind to promoters in a Ste12-independent manner (*Figure 6H*, *Figure 6—figure supplement 1D*). In line with Kar4, ChIP-seq of Ime4, Mum2, and Slz1 showed no enrichment throughout the genome during meiosis (*Figure 6G*, *Figure 6—figure supplement 1C*). As a control, we included ChIP-seq of the key meiotic transcriptional regulator Ume6 under the same conditions, which showed, as expected, enrichment at over 1000 genomic locations (*Figure 6G*; *Chia et al., 2021*). Lastly, we examined whether the Ste12 transcription factor, which is important for Kar4 recruitment during mating, plays a role in meiosis (*Kurihara et al., 1996*; *Lahav et al., 2007*). We found that *ste12Δ* did not affect the onset of meiosis (*Figure 6—figure*

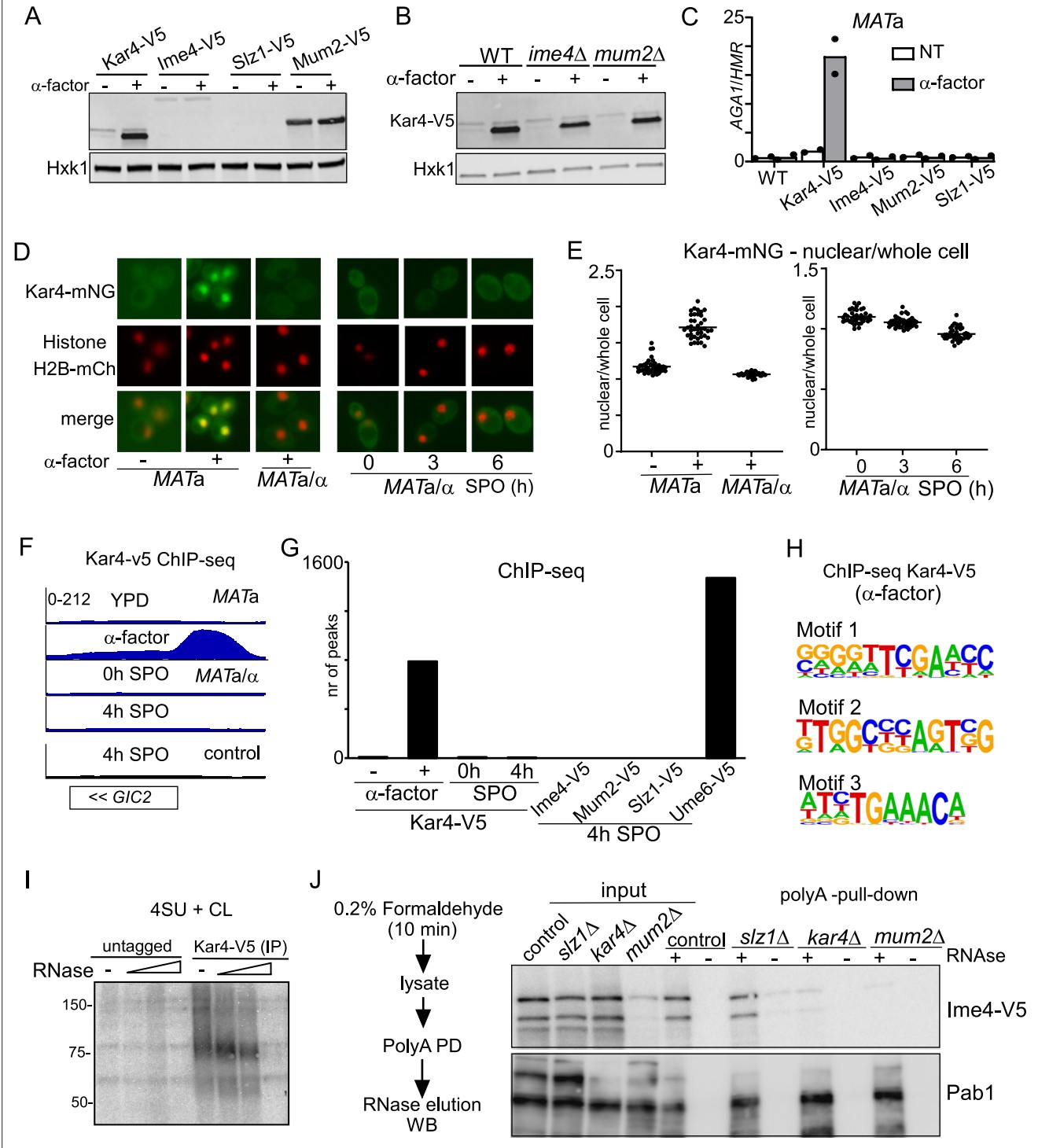

**Figure 6.** Kar4 functions during mating and meiosis differ. (**A**) Expression of Kar4, Ime4, Slz1, and Mum2, either untreated or treated with mating pheromone. *MAT*a harbouring V5 tagged alleles of Kar4, Slz1, Ime4, or Mum2 (FW8199, FW6426, FW5898, and FW6428) cells were grown in rich medium (YPD) until exponential growth, and then treated with α-factor for 30 min. Samples were collected for western blot and probed with anti-V5 antibodies. Hxk1 was used as a loading control. (**B**) Kar4 expression in WT, *ime4Δ*, or *mum2Δ MAT*a cells (FW8199, FW9415, FW8574) untreated or treated α-factor. Samples were collected for western blot and probed with anti-V5 antibodies. Hxk1 was used as a loading control. (**C**) Chromatin immunoprecipitation (ChIP) of untagged control, Kar4-V5, Ime4-V5, Mum2-V5, and Slz1-V5 *MAT*a cells (FW1509, FW8199, FW5898, FW6426, and FW6428). Cells were grown in YPD and were either not treated or treated with α factor for 2 hr. (**D**) Kar4 localization in cells in rich medium, either untreated or treated α-factor and in cells entering meiosis. Cells were induced to enter meiosis in sporulation medium (SPO), and samples were taken at 0, 3, and 6 hr in SPO. Kar4

*Figure 6 continued on next page*

*Figure 6 continued*

fused to mNeongreen (mNG) was used for the analysis. To determine nuclear Kar4-mNG signal, we used histone H2B fused to mCherry (H2B-mCh). For the analysis, we used *MAT*a and *MAT*a/α cells (FW8615 and FW8646). Representative images are shown. (**E**) Quantification of nuclear over whole cell mean signal for Kar4-mNG of data in (**C**). At least n = 50 cells were quantified for the analysis. (**F**) Chromatin immunoprecipitation followed by deep-sequencing (ChIP-seq) of *MAT*a Kar4-V5 (FW8199) untreated or treated with α factor, and *MAT*a/α Kar4-V5 (FW8616) at 0 and 4 hr in SPO as well as untagged control cells (FW1509 and FW1511). Samples were crosslinked with formaldehyde, extracts were sonicated, and protein-DNA complexes were purified using anti-V5 agarose beads and reverse crosslinked. Purified DNA was subjected to deep sequencing. ChIP-seq signals for the *GIC2* and *FUS1* loci are shown. (**G**) Quantification of the number of ChIP-seq peaks. (**H**) Motif analysis of Kar4-binding sites identified in Kar4-V5 ChIP-seq of α factor treated cells. (**I**) Photo-activatable crosslinking of Kar4. Control and Kar4 (FW1511 and FW8633) cell entering meiosis were incubated with 4thiouracil (4TU). Cells were UV-crosslinked, and Kar4 was immunoprecipitated from protein extracts. The Kar4-IP was treated with different concentrations of RNase, and subsequently radioactively labelled p32. Kar4-RNA complexes were separated by SDS-PAGE. (**J**) Chemical RNA-protein interactome analysis for assessing Ime4 binding to RNA. For the analysis, we used Ime4-V5 cells in control, *slz1Δ*, *kar4Δ*, *mum2Δ* backgrounds (FW6057, FW8362, FW8264, and FW8633). In short, cells entering meiosis (4 hr SPO) were crosslinked with formaldehyde, polyA mRNAs were pulled down from protein extracts with oligo-dT coated magnetic beads, washed, samples were eluted with RNase, and assessed by western blotting. Membranes were probed for anti-V5 to detect Ime4. As a positive control, membranes were probed with anti-Pab1.

The online version of this article includes the following source data and figure supplement(s) for figure 6:

**Source data 1.** LI-COR scan Kar4-V5 and Hxk1.

**Source data 2.** LI-COR scan Kar4-V5 and Hxk1.

**Source data 3.** Autoradiograph of Kar4-V5 IP.

**Source data 4.** Western blot of Ime4-V5.

**Source data 5.** Western blot of Pab1.

**Figure supplement 1.** Kar4 functions during mating and meiosis differ.

**Figure supplement 1—source data 1.** Western blot of Kar4-V5.

**Figure supplement 1—source data 2.** Western blot of Pab1.

**Figure supplement 1—source data 3.** Western blot of input and IP Kar4-V5.

---

supplement 1E). The above results thus rule out the requirement of additional MTC components during mating and further rule out a Kar4-mediated association of the yeast MTC with chromatin during early meiosis. We conclude that the Kar4 function in the MTC during meiosis differs from its known role as transcription regulator in the mating response pathway.

## Kar4 stabilizes the MTC on mRNAs

Given the different roles of Kar4 in meiosis and mating, and that Kar4 was not required for stabilization of Ime4, we sought to better understand the molecular role played by Kar4 in the MTC complex during meiosis. Specifically, we examined whether Kar4 was required for allowing Ime4 to associate with mRNAs.

We assessed RNA binding of the m6A writer complex in cells staged in early meiosis. We applied RNA-protein interaction capture after chemical crosslinking to purify protein-RNA complexes from cells after short and mild crosslinking with formaldehyde using oligo(dT) magnetic beads (***Figure 6— figure supplement 1F***). The assay revealed that both Ime4 and Kar4 efficiently co-purify with mRNA following RNAse elution from the beads, while pull-down from extracts pre-treated with RNAse did not show Ime4 and Kar4 signals with immunoblot as expected (***Figure 6—figure supplement 1F***). As an orthogonal approach, we performed photo-activatable UV crosslinking to determine whether RNA associate with Kar4. The analysis revealed Kar4 can crosslink to RNA (***Figure 6I***, ***Figure 6—figure supplement 1G***). Thus, Ime4 and Kar4 can both associate with mRNAs during early meiosis.

Next, we assessed whether Ime4 association with mRNAs is regulated by Kar4, Slz1, and Mum2 (***Figure 6J***). We applied RNA-protein interaction capture on extracts from *slz1Δ*, *kar4Δ*, and *mum2Δ* cells that were induced to enter meiosis. After the polyA pull-down, and elution with RNase, Ime4 was specifically enriched to comparable degrees in both the control and *slz1Δ* (***Figure 6J***). In contrast, Ime4 association to mRNAs was reduced in *kar4Δ* cells with the no effect on Ime4 levels in input. Given Mum2's stabilizing role in the yeast MTC, the Ime4 signal in *mum2Δ* cells was greatly reduced in both input and eluate from polyA pull-down. The Pab1 protein signal, which served as a positive control for RNA binding, did not differ across the different samples. These data suggest that Kar4 is important

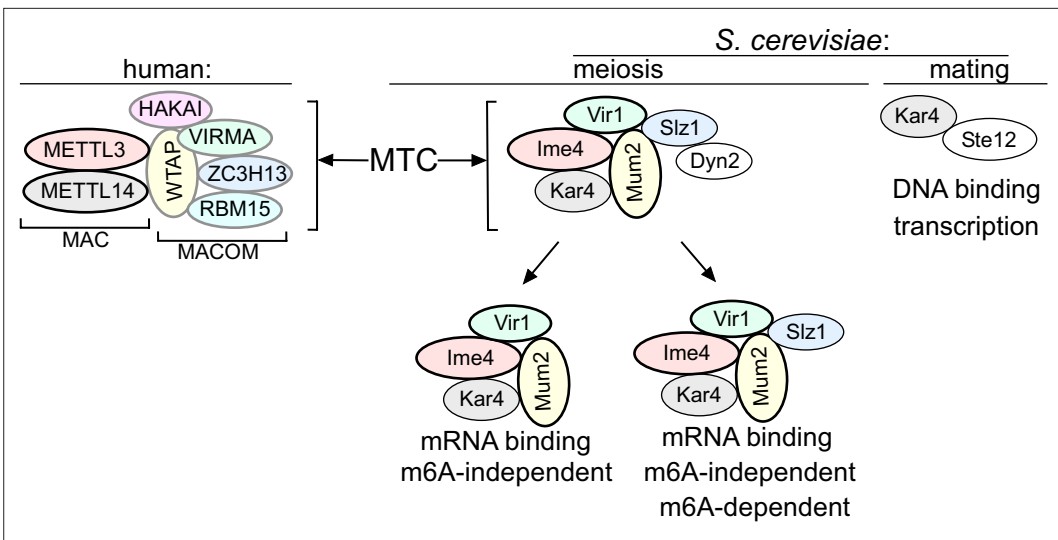

**Figure 7.** Model of the yeast methyltransferase complexes (MTC). Comparison of the yeast and the human MTCs. The human MTC consists of MAC and MACOM subcomplexes, while the yeast MTC forms a single complex. The conserved subunits are colour indicated. Mum2, Vir1, and Ime4 mutually stabilize each other. Further indicated are the m6A-dependent and m6A-independent MTC requirements, and Kar4's separate function in mating.

for stabilizing the interaction between Ime4 (and likely the whole MTC) and mRNAs, and that Slz1 is not required for this.

## Discussion

Previous work showed that the yeast MTC consists of only three subunits (Mum2, Ime4, Slz1), which contrasted the six or seven subunits described for MTCs in mammals, *Drosophila*, and plants, respectively (*Agarwala et al., 2012*). Here, we showed that the yeast MTC comprises six subunits of which five are essential for m6A deposition and have orthologues in the mammalian MTC. While the yeast MTC subunits are considerably more conserved than previously described, our findings suggest that the MTC is configured differently compared to the mammalian and *Drosophila* counterparts (see below). The yeast MTC exerts both m6A-dependent and -independent functions that are both crucial for yeast meiosis.

### Composition and features of the yeast MTC

Our analysis suggests that five subunits of the yeast MTC have orthologues in mammalian MTCs (Ime4/METTL3, Kar4/METTL14, Mum2/WTAP, Vir1/VIRMA, and potentially also Slz1/ZC3H13) (*Figure 7*). Thus, the composition of the yeast MTC is more conserved than previously described; however, we propose that the yeast MTC subunits are configured differently in a complex. The mammalian and *Drosophila* MTCs are configured into MAC (METTL3 and METTL14) and MACOM (WTAP, VIRMA, ZC3H13, HAKAI, and RBM15) complexes, of which MAC harbours the catalytic activity and MACOM is critical for stimulating MAC activity (*Knuckles et al., 2018*; *Su et al., 2022*). Consistent with the presence of two complexes, diverse studies over the years have reported that METTL3 and METTL14 mutually stabilize each other (*Śledź and Jinek, 2016*; *Wang et al., 2016*). Likewise, subunits of the MACOM complex (e.g. WTAP and VIRMA) mutually stabilize each other (*Yue et al., 2018*). However, depletion of components in one of subcomplex was not reported to impact the integrity of the other subcomplex (*Yue et al., 2018*). In yeast, we show that five subunits (Mum2, Ime4, Vir1, Kar4, and Slz1) are essential for m6A deposition and form a stable complex. We also provide evidence that the yeast MTC subunits Ime4, Mum2, and Vir1 stabilize each other and Kar4, indicating that in yeast the MAC and MACOM subunits are interweaved (*Figure 7*). This interweaved arrangement of the yeast MTC is accompanied also by loss of a mutual stabilizing interaction between Ime4 and Kar4, which is in contrast to their mammalian counterparts (*Śledź and Jinek, 2016*; *Wang et al., 2016*).

We showed that Ygl036w-Vir1 is the orthologue of Virma in mammals, Virilizer in *Drosophila*, and Vir in plants, respectively (*Haussmann et al., 2016*; *Růžička et al., 2017*; *Yue et al., 2018*). Like its mammalian/*Drosophila*/plant counterparts, Vir1 has a central role in the MTC and m6A deposition. Structurally, Vir1 and VIRMA showed strong conservation; however, Vir1 does not contain the VIRMA domain, which is thought to bridge the interaction between HAKAI and Fl2(d) in *Drosophila* (*Bawankar et al., 2021*; *Wang et al., 2021*). The absence of this domain may underlie the lack of a discernible HAKAI orthologue in yeast. We showed that in yeast Vir1 stabilizes Mum2, Ime4, and Kar4, and that it is stabilized by Mum2 and Ime4. This contrasts with the mammalian MTC, where VIRMA and WTAP stabilize each other, but not METTL3 and METTL14 (*Yue et al., 2018*).

Though Kar4 has been proposed to be the METTL14 orthologue in yeast, no molecular function in meiosis and in m6A deposition for yeast Kar4 was described previously. Here we showed that Kar4 is a critical subunit of the yeast MTC. Like METTL14, Kar4 is essential for m6A deposition and facilitates mRNA binding of the yeast MTC in vivo. We speculate that that the role of Kar4 in bridging mRNA binding of the yeast MTC is critical since in yeast the MTC likely does not harbour a designed RNA binding protein such as RBM15 (*Patil et al., 2016*). This function of Kar4 is perhaps related to how METTL14 stabilizes METTL3 on mRNAs in vitro and in vivo (*Śledź and Jinek, 2016*; *Wang et al., 2016*).

We found that Slz1 shows some sequence similarities with ZC3H13 in humans, suggesting that Slz1 and ZC3H13 are possible orthologues despite being very different in size. Previous work showed that Slz1 is required for localizing the yeast MTC to the nucleus, which is also consistent with findings for ZC3H13 in mammals where it also is important for MTC nuclear localization (*Schwartz et al., 2013*; *Wen et al., 2018*). In *Drosophila*, Flacc (ZC3H13/Slz1) bridges the interaction between Nito (RBM15) and the MTC (*Knuckles et al., 2018*). However, our analysis failed to identify a yeast orthologue for Nito (RBM15) (*Knuckles et al., 2018*). Slz1 does not seem to be important for MTC association with mRNAs and is dispensable for the m6A-independent role of the MTC. However, Slz1 is essential for driving the m6A deposition reaction since cells lacking Slz1 showed no m6A. Recent structural work on the human MTC showed that ZC3H13 facilitates a conformational change in VIRMA and WTAP, which may be important for m6A deposition (*Su et al., 2022*). The relevance of such a role in yeast remains to be confirmed.

Our analysis showed that dynein light chain protein, Dyn2, is a subunit of the yeast MTC. Despite the small size of Dyn2, we identified Dyn2 in all Mum2-IP-MS experiments. Moreover, the interaction between Dyn2 and the yeast MTC Mum2 is RNA independent but requires Slz1. In contrast to the remaining components, loss of Dyn2 led to a partial loss of mRNA methylation accompanied by a very mild meiotic phenotype. We therefore decided to primarily focus on the remaining components in this study. However, it is worth noting that in *Caenorhabditis elegans* a dynein light chain protein Dlc1 is involved in stabilizing the m6A writer Mett10 (*Dorsett and Schedl, 2009*). More work is needed to dissect the role of Dyn2 in the yeast MTC.

## Multifunctionality of the yeast MTC

Previous studies have pointed at m6A-dependent and m6A-independent roles played by Ime4 (*Agarwala et al., 2012*; *Clancy et al., 2002*). These conclusions were primarily guided by the differences in phenotypes between *IME4* deletion and an *IME4* catalytic inactive mutant, both of which abolish methylation but are associated with phenotypes of distinct severity. The nature of such m6A-independent roles had been so far unknown. Such a phenotype could be 'MTC complex independent,' that is, mediated via a completely new function of ime4, unrelated not only to its catalytic activity but also to its participation in the methylation complex. Alternatively, such a phenotype could be MTC-dependent. The discovery of additional components of the yeast MTC allowed us to uncover that the substantially more severe phenotype of the *IME4* deletion mutant was shared also upon loss of other components of the yeast MTC, strongly suggestive of an MTC-dependent function. These results suggest that an intact MTC is critical for yeast meiosis, independent of its ability to methylate RNA. In this sense, these results are reminiscent of studies conducted on diverse proteins playing a role in modifying the ribosomal RNA (rRNA) or transfer RNA (tRNA), often revealing that catalytically inactive mutations display considerably less pronounced phenotypes than full deletions (*Létoquart et al., 2014*; *Leulliot et al., 2008*; *Sharma et al., 2015*; *Sharma et al., 2013*; *White et al., 2008*). In such cases, there was no moonlighting activity of the modifying enzyme reported, but instead the

modifying enzyme played a scaffolding or chaperoning role, for which its catalytic activity was not essential.

m6A-independent functions have been reported for METTL3 and METTL14 of the mammalian MTC; however, to our knowledge no role for an inactive but intact MTC in mammals has been documented so far (*Barbieri et al., 2017*; *Liu et al., 2021*). In addition to the m6A-dependent and -independent roles of the yeast MTC in meiosis, Kar4 also has a separate function where it acts as a transcription factor in mating (*Kurihara et al., 1996*; *Lahav et al., 2007*). The functions in transcription and in the MTC of Kar4 are not linked since Kar4 does not stably associate with chromatin during meiosis.

At the time of preparation of this article, a related study showed that Ylg036w-Vir1 is important for m6A deposition and meiosis in yeast (*Park et al., 2023a*). Additionally, the same group showed in two different studies that Kar4 has an important role in controlling gene expression in meiosis, and elegantly identified separation of function mutants that affect either the mating or meiotic function of Kar4 (*Park et al., 2023b*; *Park et al., 2023c*). Both works support the findings described in our article.

## Concluding remarks

In conclusion, we describe a new composition of the yeast MTC. We showed that yeast MTC components are considerably more conserved than previously described, and yet the yeast MTC is re-configured compared to mammalian and *Drosophila* MTCs. The yeast MTC exerts both m6A-dependent and m6A-independent functions that are both important for meiosis. Our findings expand the relevance of yeast as a model system for understanding the molecular mechanisms regulating m6A deposition, as well as other m6A-independent roles of MTCs.

## Methods

### Plasmids and yeast strains

For this study, we used strains derived from the sporulation proficient SK1 background and experiments were carried out in diploid cells unless stated otherwise. Gene deletions and carboxy terminal tagging with mNeongreen, 3xV5, TEV Protein A, or AID were performed using one-step gene replacement protocol described previously (*Longtine et al., 1998*; *Tam and van Werven, 2020*). Depletions using C-terminal auxin-inducible degron (AID) tag were performed as described by *Nishimura et al., 2009*. A short version of the AID-tag, known as mini-AID, was used for depleting Slz1, Pab1, and Npl3 (*Morawska and Ulrich, 2013*). To induce depletion during meiosis, we used copper-inducible *Oryza sativa* TIR (*osTIR*) ubiquitin ligase under the control of the *CUP1* promoter (*Varier et al., 2022*). The strain genotypes are listed in *Supplementary file 3*, the plasmids used in *Supplementary file 4*, and the oligo sequences in *Supplementary file 5*.

### Growth conditions

Cells were grown in YPD (1.0% (w/v) yeast extract, 2.0% (w/v) peptone, 2.0% (w/v) glucose, and supplemented with uracil (2.5 mg/l) and adenine (1.25 mg/l)). To induce meiosis, a standard protocol for sporulation was followed as described previously (*Varier et al., 2022*). In short, cells were grown until saturation for 24 hr in YPD, then grown in pre-sporulation medium BYTA (1.0% [w/v] yeast extract, 2.0% [w/v] bacto tryptone, 1.0% [w/v] potassium acetate, 50 mM potassium phthalate) for about 16 hr and shifted to sporulation medium (SPO) (0.3% [w/v] potassium acetate and 0.02% [w/v] raffinose). All experiments were performed at 30°C in a shaker incubator at 300 rpm. For mating experiments, cells were grown to early log phase in YPD, when 1 uM α-factor was added, and cells were incubated with the pheromone for 30 min before collection. To enable efficient depletion of mini-AID strains (Slz1, Vir1, Pab1, and Npl3), 1 mM of indole-3-acetic acid (IAA) and 50 µM of CuSO$_4$ were added 2 hr after cells were shifted to SPO and samples collected at 4 hr in SPO. As a control, the same volume of dimethyl sulphoxide (DMSO) was added to the yeast culture.

### RNA extraction

RNA extraction was performed from yeast pellets using acid phenol:chloroform pH 4.5 and Tris-EDTA-SDS (TES) buffer (0.01 M Tris–HCl pH 7.5, 0.01 M EDTA, 0.5% w/v SDS). Samples were further treated with rDNase (Cat# 740.963, Macherey-Nagel) and column purified (Cat# 740.948, Macherey-Nagel).

## RT-qPCR

For reverse transcription, ProtoScript II First Strand cDNA Synthesis Kit (New England Biolabs) was used and 500 ng of total RNA was provided as template in each reaction. qPCR reactions were prepared using Fast SYBR Green Master Mix (Thermo Fisher Scientific) and transcript levels were quantified from the cDNA on Quantstudio 7 Flex Real Time PCR instrument. Signals were normalized over *ACT1*.

## DAPI counting

Cells were collected from sporulation cultures, pelleted via centrifugation, and fixed in 80% (v/v) ethanol for a minimum of 2 hr at 4°C. The cells were resuspended in PBS with 1 μg/ml DAPI. The proportion of cells containing two or more nuclei were considered meiosis.

## ChIP and ChIP-seq

Chromatin immunoprecipitation (ChIP) was performed as described previously (*Moretto et al., 2018*). In short, cells were fixed in 1.0% v/v formaldehyde. Cells were lysed in FA lysis buffer (50 mM HEPES–KOH, pH 7.5, 150 mM NaCl, 1 mM EDTA, 1% Triton X-100, 0.1% Na-deoxycholate, 0.1% SDS, and protease cocktail inhibitor [cOmplete mini EDTA-free, Roche]) and chromatin was sheared by sonication using a Bioruptor (Diagenode, nine cycles of 30 s on/off). Extracts were incubated for 2 hr at room temperature with anti-V5 agarose beads (Sigma), washed twice with FA lysis buffer, twice with wash buffer 1 (FA lysis buffer containing 0.5 M NaCl), and twice with wash buffer 2 (10 mM Tris–HCl, pH 8.0, 0.25 M LiCl, 1 mM EDTA, 0.5% NP-40, 0.5% Na-deoxycholate). Reverse cross-linking was done in 1% SDS-TE buffer + Ribonuclease A (10 ng/μl) (100 mM Tris pH 8.0, 10 mM EDTA, 1.0% v/v SDS) at 65°C overnight. Proteinase K-treated samples were column purified.

qPCR was used for determining the association with AGA1 promoter, and HMR was used as a background control. For ChIP-seq, libraries were prepared using the KAPA Hyperprep kit (Roche) according to the manufacturer's protocol. Libraries were size-selected for DNA fragments between 150 nt and 700 nt using gel extraction. Purified libraries were further quantified and inspected on a Tapestation (Agilent Technologies) and sequenced on an Illumina HiSeq 2500 to an equivalent of 50 bases single-end reads, at a depth of approximately 16 million reads per library.

## ChIP-seq data analysis

For ChIP-seq, data adapter trimming was performed with cutadapt (version 1.9.1) with parameters '--minimum-length=25 --quality-cutoff=20 -a AGATCGGAAGAGC,' while for Ndt80 ChIP-seq, the cutadapt parameters are '-a AGATCGGAAGAGC --minimum-length = 20.' BWA (version 0.5.9-r16) with default parameters was used to perform genome-wide mapping of the adapter-trimmed reads to the SK1 genome (*Li et al., 2009*; *Martin, 2011*). Duplicate marking was performed using the picard tool MarkDuplicates (version 2.1.1; http://broadinstitute.github.io/picard). Further filtering was performed to exclude reads that were duplicates and ambiguously mapped.

The peak calling for ChIP-seq data was done using 'macs2 callpeak' in MACS2 with parameters '-g 12000000 -m 3 100 -B -q 0.05' (*Liu, 2014*). Based on the output of 'macs2 callpeak,' the signal tracks were generated by the 'macs2 bdgcmp' command with parameters '-m ppois' (*Heinz et al., 2010*; *Liu, 2014*; *Quinlan and Hall, 2010*).

## Western blotting and quantification

3.6 ODs of yeast cells were pelleted from cultures via centrifugation for protein expression analysis. Proteins were precipitated from whole cells via trichloroacetic acid (TCA). Cells were lysed with glass beads using a bead beater and lysis buffer (50 mM Tris, 1 mM EDTA, and 2.75 mM DTT). Proteins were denatured in 3× sample buffer (9% [w/v] SDS and 6% [v/v] β-mercaptoethanol) at 100°C and separated by SDS/PAGE (4–12% gels). A PVDF membrane was used for protein transfer. Blocking was performed using 5% (w/v) dry skimmed milk. Proteins were detected using an ECL detection kit or using infrared fluorescent antibodies visualized with LI-COR CLx. Samples were normalized using Hxk1 as a loading control. Images were taken using the LI-COR (manufacturer) and analysed using the Image Studio Lite software.

For the chemical RNA interactome capture (RIC) experiments, 15 μg of total protein (input control) and 20% of eluates were resolved on 4–20% SDS polyacrylamide gels (SDS-PAGE) and transferred to

nitrocellulose (NC) membranes (Cytiva Life Sciences, 10600003). Membranes were blocked in PBS-0.1% Tween-20 containing 5% skimmed milk, probed with designated antibodies and horseradish peroxidase (HRP)-coupled secondary antibodies, and developed with the Immobilon Western Chemi-luminescent HRP Substrate (MerckMillipore, WBKLS0500). Blots were recorded with an Amersham A600 gel documentation system (Cytiva Life Sciences). The following antibodies were used: V5 tag monoclonal antibody (1:2000; Thermo Fisher, R96025), mouse anti-Pab1 (1:5000; Antibodies online, ABIN1580454), mouse anti-Act1 (1:2500; MP Biomedicals, 0869100), and HRP-conjugated sheep anti-mouse IgG (1:5000; Cytiva, NA931V).

## Immunoprecipitation and mass spectrometry

Yeast cells containing the Mum2-TEV-ProA-tagged protein and control cells were induced to undergo sporulation as previously described (*Spedale et al., 2010*). For each immunoprecipitation experiment, we used 1250 OD units of yeast cells. Cells were resuspended in lysis Buffer E (10 mM HEPES-NaOH, pH 8.0, 150 mM NaCl, 0.1% Tween-20, 10% glycerol, 1 mM dithiothreitol [DTT]) supplemented with RNase inhibitors (HALT protease inhibitor), and snap frozen, and processed into a powder using a Spex SamplePrep Freezer/Mill Cryogenic Grinder. Cell powder was dissolved in 12 ml Lysis Buffer E supplemented with RNase inhibitors and rotated to mix for at least 30 min in 4°C. Large debris was removed by a short spin (3000 × $g$ for 2 min, 4°C), and to clarify the lysate 45.000 rpm for 1 hr. All handling of the lysate was done at 4°C. For each sample, 250 µl Streptavidin M-280 Dynabeads (Thermo Fisher Scientific) coated with 30 µl biotinylated anti-rabbit IgG (Vector Laboratories) was added. The yeast extracts were incubated with the beads for 2 hr at 4°C. For samples treated with RNase I, the protein concentration was determined with Bradford reagent (manufacturer) and 1 U RNase I per mg protein was added at the start of the incubation. Beads were collected using magnetic stands, washed once with lysis buffer E, and washed three times with Cleavage Buffer (10 mM Tris–HCl, pH 8.0, 150 mM NaCl, 0.5 mM EDTA, 0.1% Tween-20, 1 mM DTT). Finally, beads were resuspended in 49 µl cleavage buffer. Proteins were eluted by the addition of 1 uL TEV Protease (New England Biolabs) and incubated for 2 hr at room temperature, gently shaking at 450 rpm.

Reduced and alkylated proteins were prepared by in-gel digesting overnight at 37°C using 100 ng trypsin (Promega). The supernatant was dried by vacuum centrifugation, and the samples were resuspended in 0.1% trifluoro acetic acid. 1–10 µl of acidified protein digest was loaded on an Ultimate 3000 nanoRSLC HPLC (Thermo Scientific) onto a 20 mm × 75 um Pepmap C18 trap column (Thermo Scientific) prior to elution via a 50 cm × 75 um EasySpray C18 column into a Lumos Tribrid Orbitrap mass spectrometer (Thermo Scientific). A 70 min gradient of 6–40% B was used followed by washing and re-equilibration (A = 2% ACN, 0.1% formic acid; B = 80% ACN, 0.1% formic acid). The Orbitrap was operated in 'Data Dependent Acquisition' mode followed by MS/MS in 'TopS' mode using the vendor-supplied 'universal method' with default parameters. Raw files were processed using Maxquant (maxquant.org) and Perseus (maxquant.net/perseus) with a recent download of the UniProt *Saccharomyces cerevisiae* reference proteome database and a common contaminants database. A decoy database of reversed sequences was used to filter false positives, at a peptide false detection rate of 1%. $t$-tests were performed with a permutation-based FDR of 5% to address multiple hypothesis testing.

## Fluorescence microscopy and image quantification

Kar4-mNeongreen and Htb2-mCherry image acquisition was conducted using a Nikon Eclipse Ti inverted microscope. Exposure times were set as follows: 500 ms GFP, and 50 ms mCherry. An ORCA-FLASH 4.0 camera (Hamamatsu) and NIS-Elements AR software (Nikon) were used to collect images. Quantification of fluorescence signals was performed using ImageJ (*Schindelin et al., 2012*). ROIs were manually drawn around the periphery of each cell or around the nucleus. The mean intensity in each channel per cell was determined and used for the analysis. Values for nuclear protein localisation were derived via the division of nuclear/whole cell signal. For the analyses, 50 cells were quantified per sample.

## m6A-mRNA quantification by LC-MS/MS and m6A-ELISA

RNA was extracted and DNase treated as described above and polyA RNA isolated after purification twice with oligodT dynabeads (Ambion, 61005) according to the manufacturer's instructions. M6A LC-MS was described previously (*Varier et al., 2022*). In short, polyA selected RNA was digested with

addition of reaction buffer (2.5 mM $ZnCl_2$, 25 mM NaCl, 10 mM Na acetate), two units of nuclease P1 (Sigma, N8630), and incubated for 4 hr at 37°C. Followed by 2 hr with 100 mM ammonium bicarbonate, 1 µl alkaline phosphatase (NEB, M0525). Formic acid was added to the reaction mixture and filtered with a 1.5 ml microfuge tube. The sample was then injected (20 µl) and analysed by LC-MS/MS using a reverse-phase liquid chromatography C18 column and a triple quadrupole mass analyser (Agilent 6470 or Thermo Scientific TSQ Quantiva) instrument in positive electrospray ionisation mode. Flow rate was at 0.2 ml/min and column temperature 25°C with the following gradient: 2 min 98% eluent A (0.1% formic acid and 10 mM ammonium formate in water) and 2% eluent B (0.1% formic acid and 10 mM ammonium formate in MeOH), 75% A and 25% B up to 10 min, 20% A and 80% B up to 15 min, 98% A and 2% B up to 22.5 min. Nucleosides were quantified using a calibration curve of pure nucleosides standards.

For ELISA, we either used the EpiQuik m6A RNA methylation quantification kit from EpiGentek (P-9005) according to the manufacturer's protocol or an m6A ELISA protocol previous described (*Ensinck et al., 2023*).

## Chemical crosslinking RNA-protein interactome capture

To determine Ime4 and Kar4 binding to mRNAs, we adapted an RNA-protein interactome protocol using chemical crosslinking instead of UV crosslinking (*Matia-González et al., 2021*; *Na et al., 2021*; *Patton et al., 2020*). In short, RNA-protein complexes were crosslinked chemically using 0.2% formaldehyde (Thermo Fisher, 28908) for 10 min at 30°C. Crosslinking was quenched by adding 2.5 M glycine to a final concentration 125 mM followed by incubation for 5 min in an orbital shaker at 30°C. Cells were collected by centrifugation, washed twice with PBS, then flash-frozen in liquid nitrogen. Cells were ground in liquid nitrogen using pre-chilled mortar and pestle. Cell powder was resuspended in 3.5 ml of RIC lysis buffer (20 mM Tris–HCl pH 7.4, 600 mM LiCl, 0.5% LiDS, 2 mM EDTA, 5 mM DTT, 1% Triton X-100, 1.5×Halt Protease Inhibitor Cocktail [Thermo Fisher, 78429], 1 mM AEBSF, and 0.2 U SUPERase RNase inhibitor [Invitrogen, AM2696]). Lysates were clarified by centrifugation. Total protein quantification was estimated with Bradford assay.

RNA-protein complexes were isolated with oligo[dT]$_{25}$ magnetic beads (New England Biolabs, S1419S). Briefly, 350 µl of oligo[dT]$_{25}$-coupled beads were pre-equilibrated by washing them five times with 1 ml of RIC lysis buffer. 3.5 mg of total protein extracts from each culture were mixed with beads, vortexed briefly then incubated at room temperature for 15 min with gentle rotation in a rotator wheel. Beads were washed twice with 1 ml of wash buffer A (20 mM Tris–HCl pH 7.5, 600 mM LiCl, 0.2% LiDS, 0.5 mM EDTA, 0.1% Triton X100, 1 × 1.5 × Halt Protease Inhibitor Cocktail, 1 mM AEBSF, and 0.2 U SUPERase), and twice with 1 ml of wash buffer B (20 mM Tris–HCl pH 7.5, 600 mM LiCl, 0.5 mM EDTA, 1 × 1.5 × Halt Protease Inhibitor Cocktail, 1 mM AEBSF, and 0.2 U ml SUPERase) each for 15 s on ice. Beads were resuspended in 60 µl elution buffer (10 mM Tris–HCl pH 7.5, 10 U of RNase I, 4 µg RNase A) and incubated at 37°C for 10 min. For mock elution control, beads were resuspended in elution buffer without RNases.

## Sequence alignment

Whole proteome sequences were acquired from UniProt (*UniProt, 2023*). Global and local sequence alignment scores were determined using EMBOSS needle and matcher, respectively, with the emboss/6.6.0 package (*Rice et al., 2000*; *Madeira et al., 2019*). Parameters employed for needle and water were EBLOSUM62 matrix with Gap_penalty: 10.0 and Extend_penalty: 0.5. Parameters for matcher were Gap_penalty: 14 and Extend_penalty: 4.

## Structural alignment

Structural predictions were downloaded from the AlphaFold database (*Jumper et al., 2021*; *Varadi et al., 2022*). Structural similarity was calculated using the TM-align algorithm (*Zhang and Skolnick, 2005*) using TM-align/20210224. The resultant TM-score was normalized by the length of the shared query protein (e.g. when comparing *YGL036W* to the whole human genome, *YGL036W* length was used).

## m6A-seq2

m6A levels were determined by m6A-seq2, as previously described (*Dierks et al., 2021*; *Schwartz et al., 2013*). Briefly, RNA was poly-A selected twice using Dynabeads mRNA DIRECT kit, and

fragmented to ~150 bp fragments. 3′ RNA barcode adaptors were then ligated (see *Supplementary file 6*), and all samples were pooled for m6A-IP-based enrichment. 90% of the mRNA material was subjected to two rounds of immunoprecipitation using two distinct anti-m6A antibodies, as follows: the RNA was first incubated for 4 hr with protein G beads (Invitrogen) coupled to a Synaptic Systems polyclonal m6A antibody (Cat# 7945). Next, it was incubated overnight with protein A beads (Invitrogen) coupled to a Cell Signaling polyclonal m6A antibody (Cat# D9D9W). RNA from input and IP libraries were reverse transcribed and PCR amplified as previously described (*Dierks et al., 2021*). cDNA libraries were quantified using Qubit RNA HS kit (Life Technologies) and Tapestation High Sensitivity D1000 ScreenTape (Agilent Technologies), pooled with a ratio of ⅓ Input to ⅔ IP samples, and sequenced.

## m6A-seq2 data analysis

Paired-end reads were first demultiplexed according to barcodes in Read2 position 4–10 using an in-house Python script. Reads were then aligned using STAR/2.5.3a to the SK1 reference genome (*Schwartz et al., 2013*) with the following additional parameters (--bamRemoveDuplicatesType UniqueIdentical, --outSAMtype BAM SortedByCoordinate,--alignIntronMax 500, --alignEndsType Local). Read were allocated to genes using txtools version 0.0.7.4 according to an SK1 gene annotation table. m6A sample index was calculated as the ratio of reads coverage between IP and Input samples, in a 51 bases window around a set of 1308 previously well-defined m6A sites (*Supplementary file 6*). To ensure adequate quantification, only sites with an average of 5 reads per base in the input fraction were used.

## RNA-seq

RNA levels were quantified by sequencing 3′ RNA ends as previously described (*Chapal et al., 2019*). Briefly, RNA was reverse transcribed using oligo dT primers coupled with internal barcodes. RNA-DNA hybrids were then pooled and tagmented using Tn5, adding adaptors to their 5′ end, followed by PCR amplification of the fragments, inserting Illumina barcodes and adaptors. A total of four biological replicates were sequenced for each mutant-time point combination, two of each were processed in each of two pools.

## RNA-seq data analysis

Paired-end reads were demultiplexed according to barcodes in Read2 position 1–6 using an in-house Python script. Read1 only were then aligned using STAR/2.5.3a (*Dobin et al., 2013*), with the following additional parameters (--outSAMtype BAM Unsorted, --alignIntronMax 500,--alignEndsType Local, --outBAMsortingBinsN 300, --limitBAMsortRAM 1254135158) and then deduplicated using UMI-tools 1.0.1. Reads were allocated to a custom-built transcriptome annotation of 3′ UTRs +–50 bp, using txtools 0.0.7.4. Transcription end site (TES) coordinates for SK1 cells entering meiosis were previously described (*Chia et al., 2017*). For a subset of transcripts, the UTR length was used to determine the end of transcripts (*Nagalakshmi et al., 2008*). Transcripts 3′ ends were annotated as ending at the farthest TES from the coding sequence and starting either 600 bases upstream of the closest TES or at the 5′ UTR start. The TES coordinates used are listed in *Supplementary file 7*. Samples with more than 2 standard deviations below the mean mapped reads per sample or with low gene coverage (below 5000 genes mapped) were not included in the analysis (this excluded three samples).

Counts were normalized using DESEQ2/1.34.0 (*Love et al., 2014*). PCA was done using PCAtools with removeVar = 0.1 option. Differentially expressed genes were determined by comparing mutated strain samples with their WT counterparts at matching time points, with the thresholds of adjusted p-value of 0.01 and an absolute log2 (FoldChange) of 1.5. Meiosis gene clusters were taken from a previously described dataset (*Chu et al., 1998*). *Supplementary file 2* contains the RNA-seq data normalized readcounts.

## Cloning, expression, and purification of recombinant yeast MTC

Ime4, Mum2, Slz1, Kar4, and Vir1 genes were codon-optimized for *Escherichia coli* expression, synthesized de novo, and cloned into pACEBac1 (Epoch Life Sciences). During gene synthesis, a StrepII tag (SII) and site for cleavage by 3C PreScission protease were inserted at the N-terminal domain of Ime4 (SII-3C-Ime4). pACEBac1 plasmids were amplified by PCR using the original biGBac primers

and introduced into pBIG1a by Gibson assembly using a modified version of the biGBac system (*Hill et al., 2019*). The final construct carrying the five subunits was verified using full-plasmid sequencing. Bacmid DNA was isolated from *E. coli* DH10 EmBacY cells, as described (Bieniossek, 2008 #365; *Weissmann et al., 2016*). To make P1 virus, six-well dishes were seeded with $1.0 \times 10^6$ *Sf9* cells per well in 2.0 ml InsectExpress medium (Lonza). Cells were transfected with 20 ml of bacmid per well, using Cellfectin II reagent as described by the manufacturer (Gibco). The supernatant (P1 virus) was harvested 72 hr post-transfection, adding 40 ml of newborn calf serum (Gibco) before storage at 4°C. P2 (amplified P1) virus was prepared by infecting suspension cultures of *Sf9* cells at $5.0 \times 10^5$/ml with 10% v/v P1 virus and incubating for 4–5 d (130 rpm, 27°C). Cells were checked for fluorescence, harvested by centrifugation ($1000 \times g$, 5 min), and the supernatant collected and stored at 4°C. For further viral amplification, P3 (amplified P2) was prepared infecting Sf9 cells at $5.0 \times 10^5$ cells/ml with 10% v/v P2 virus and incubating for 48 hr before harvesting the virus. Large-scale expression cultures were then set up by infecting 2–6 L suspension cultures of *High5* insect cells at $5.0 \times 10^5$/ml with 0.5% v/v P3 virus. Following incubation (130 rpm, 27°C), cells were harvested 48 hr post-infection by centrifugation ($1000 \times g$, 20 min, 4°C), and stored at –80°C.

Cell pellets from 2 L *High5* cells were resuspended in 50 mM HEPES pH 7.9, 300 mM NaCl, 1 mM PMSF, 10% glycerol, 1 mM TCEP, supplemented with 5 U/ml benzonase and EDTA-free protease inhibitors, and lysed by homogenization. The lysate was cleared by centrifugation (12,000 rpm, 30 min, 4°C), 0.45 µm filtered, and loaded into a 5 ml StrepTrap HP column (Cytiva) pre-equilibrated in 50 mM HEPES pH 7.9, 300 mM NaCl, 1 mM PMSF, 1 mM TCEP. Elution was performed using the pre-equilibration buffer supplemented with 50 mM biotin. The purified complex was used immediately for SDS-PAGE analysis, and the sample was stored at 4°C.

## Statistical analyses
Details of statistical tests used, sample number, and number of independent experiments are included in the relevant figure legends.

## Acknowledgements
We thank the members of the van Werven lab for critical reading of the manuscript. This research was funded in whole, or in part, by the Wellcome Trust (FC001203). For the purpose of Open Access, the author has applied a CC BY public copyright licence to any Author Accepted Manuscript version arising from this submission. This work was supported by the Francis Crick Institute (FC001203), which receives its core funding from Cancer Research UK (FC001203), the UK Medical Research Council (FC001203), and the Wellcome Trust (FC001203). SS received funding by the Israel Science Foundation (913/21) and by the European Research Council (ERC) under the European Union's Horizon 2020 research and innovation programme (grant nos. 714023 and 101000970).

## Additional information

### Funding

| Funder | Grant reference number | Author |
| --- | --- | --- |
| Wellcome Trust | FC001203 | Imke Ensinck<br>Folkert J van Werven<br>Theodora Sideri<br>Waleed S Albihlal |
| Cancer Research UK | FC001203 | Imke Ensinck<br>Folkert J van Werven<br>Theodora Sideri<br>Waleed S Albihlal |
| Medical Research Council | FC001203 | Imke Ensinck<br>Folkert J van Werven<br>Theodora Sideri<br>Waleed S Albihlal |

| Funder | Grant reference number | Author |
|---|---|---|
| Israel Science Foundation | 913/21 | Alexander Maman Schraga Schwartz |
| European Research Council | 714023 | Alexander Maman Schraga Schwartz |
| European Research Council | 101000970 | Alexander Maman Schraga Schwartz |

The funders had no role in study design, data collection and interpretation, or the decision to submit the work for publication. For the purpose of Open Access, the authors have applied a CC BY public copyright license to any Author Accepted Manuscript version arising from this submission.

## Author contributions

Imke Ensinck, Conceptualization, Data curation, Formal analysis, Validation, Investigation, Visualization, Methodology, Writing – original draft, Project administration, Writing – review and editing; Alexander Maman, Conceptualization, Resources, Data curation, Formal analysis, Validation, Investigation, Visualization, Methodology, Writing – original draft, Writing – review and editing; Waleed S Albihlal, Data curation, Formal analysis, Validation, Methodology; Michelangelo Lassandro, Data curation, Formal analysis, Investigation, Methodology; Giulia Salzano, Theodora Sideri, Steven A Howell, Enrica Calvani, Harshil Patel, Data curation, Formal analysis, Methodology; Guy Bushkin, Investigation; Markus Ralser, Supervision, Funding acquisition; Ambrosius P Snijders, Mark Skehel, Formal analysis, Supervision, Methodology; Ana Casañal, Data curation, Formal analysis, Supervision, Methodology, Project administration; Schraga Schwartz, Conceptualization, Resources, Data curation, Formal analysis, Supervision, Funding acquisition, Investigation, Visualization, Methodology, Writing – original draft, Project administration, Writing – review and editing; Folkert J van Werven, Conceptualization, Data curation, Formal analysis, Supervision, Funding acquisition, Investigation, Visualization, Methodology, Writing – original draft, Project administration, Writing – review and editing

## Author ORCIDs

Imke Ensinck ⓘ http://orcid.org/0000-0001-8468-9199
Waleed S Albihlal ⓘ http://orcid.org/0000-0002-2025-9787
Schraga Schwartz ⓘ http://orcid.org/0000-0002-3671-9709
Folkert J van Werven ⓘ http://orcid.org/0000-0002-6685-2084

Reviewer #1 (Public Review): https://doi.org/10.7554/eLife.87860.3.sa1
Reviewer #2 (Public Review): https://doi.org/10.7554/eLife.87860.3.sa2
Author Response https://doi.org/10.7554/eLife.87860.3.sa3

# Additional files

## Supplementary files

- Supplementary file 1. MS data.
- Supplementary file 2. RNA-seq data.
- Supplementary file 3. Genotypes of strains used.
- Supplementary file 4. Oligo nucleotide sequences used.
- Supplementary file 5. Plasmids used.
- Supplementary file 6. List of annotated m6A sites used for m6Aseq analysis.
- Supplementary file 7. TES coordinates used for RNA-seq.
- MDAR checklist

## Data availability

The GEO accession numbers for the m6A-seq2, RNA-seq, and ChIP-seq data reported in this manuscript are: GSE224637 and GSE224836.

The following datasets were generated:

| Author(s) | Year | Dataset title | Dataset URL | Database and Identifier |
|---|---|---|---|---|
| Alexander M | 2023 | The yeast RNA methylation complex consists of conserved yet reconfigured components with m6A-dependent and independent roles | https://www.ncbi.nlm.nih.gov/geo/query/acc.cgi?acc=GSE224836 | NCBI Gene Expression Omnibus, GSE224836 |
| Ensinck E, Patel H, Van Werven FJ | 2023 | The yeast RNA methylation complex consists of conserved yet reconfigured components with m6A-dependent and independent roles | https://www.ncbi.nlm.nih.gov/geo/query/acc.cgi?acc=GSE224637 | NCBI Gene Expression Omnibus, GSE224637 |

The following previously published dataset was used:

| Author(s) | Year | Dataset title | Dataset URL | Database and Identifier |
|---|---|---|---|---|
| Varier RA, Sideri T, Capitanchik C, Manova Z | 2022 | m6A reader Pho92 is recruited co-transcriptionally and couples translation efficacy to mRNA decay to promote meiotic fitness in yeast | https://www.ncbi.nlm.nih.gov/geo/query/acc.cgi?acc=GSE193561 | NCBI Gene Expression Omnibus, GSE193561 |

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
