## [Editor Report · eLife assessment]

This **fundamental** study identifies the components of the N6-methyladenosine methyltransferase complexes in yeasts, with major differences with the same complexes in mammals and flies. The evidence supporting the conclusions is **convincing**, with rigorous high-throughput sequencing approaches and detailed functional analysis. This work will be of broad interest to colleagues in the RNA modification and meiosis fields.

---

## [Referee Report · Reviewer #1 (Public Review)]

Here, Ensinck et al. investigated the composition of the yeast mRNA m6A methyltransferase complex required for meiosis. This complex was known to contain three proteins, but is much more complex in mammals, insects and plants. Through IP-MS analysis they identified three more proteins Kar4, Ygl036w and Dyn2. Of these Kar4 and Ygl036w are homologous to Mettl14 and Virma, respectively, and, like the previously described factors are essential for m6A deposition, mating and binding of the reader Pho92 to mRNA during meiosis by evidence acquired with appropriate methodology. Dyn2 is a novel factor not described for any m6A complex and is not essential for m6A deposition, mating and binding of the reader Pho92 to mRNA during meiosis.

In addition, detailed analysis of the Slz1 revealed homology to the mammalian factor m6A complex member ZC3H13 to comprise a conserved complex of five proteins, Mettl3, Mettl14, Mum2/WTAP, Virma and Slz/ZC3H13. When co-expressed in insects cells, they co-purify stoichiometrically and presence of Mum2 as a dimer is also indicated as shown for WTAP.

Complementary to these data they show that stability of the individual complex members is affected in mutants supporting that they are stabilized through complex formation.

Furthermore, the authors then show that kar4 has additional roles in mating that is separable from its role through the m6A complex in meiosis.

The authors employ appropriate methodology throughout to address their aims and present convincing evidence for their claims. The evidence presented here reinforces that the m6A complex is evolutionary highly conserved which has broad scope for its functional analysis in humans and model organisms.

---

## [Referee Report · Reviewer #2 (Public Review)]

N6-methyladenosine (m6A), the most abundant mRNA modification, is deposited by the m6A methyltransferase complexes (MTC). While MTC in mammals/flies/plants consists of at least six subunits, yeast MTC was known to contain only three proteins. Ensinck, Maman, et al. revisited this question using a proteomic approach and uncovered three new yeast MTC components, Kar4/Ygl036w/Dyn2. By applying sequence and structure comparisons, they identified Kar4, Ygl036w, Slz1 as homologs of the mammalian METTL14, VIRMA. ZC3H13, respectively. While these proteins are essential for m6A deposition, the dynein light chain protein, Dyn2, is not involved in mRNA methylation. Interestingly, while mammalian and fly MTCs are configured as MAC (METTL3 and METTL14) and MACOM (other subunits) complexes, yeast MTC subunits appear to have different configurations. Finally, Kar4 has a different role as transcription regulator in mating, which is not mediated by other MTC members. These data establish fundamental framework for the yeast MTC and also provide novel insights for those studying m6A deposition.

---

## [Author Response]

The following is the authors’ response to the original reviews.

**Reviewer #1 (Recommendations For The Authors):**
This study is well presented and contains all the necessary experiments to support their claims. They made the interesting finding of an additional factor Dyn2. However, it is unclear whether it is present in the human complex. Hence, it would be interesting to see whether Dyn2 co-purifies when expressed with the other complex components in insect cells. Also, purification of a tagged complex from yeast would have indicated whether Dyn2 is part of the complex and whether other factors, like RBM15 or Hakai, present in humans are also present in yeast.

We agree that Dyn2 subunit is an exciting new finding that is worth further investigation. The IP-MS experiments suggest that Dyn2 is subunit of the complex and that the Dyn2 interaction is mediated via Slz1. We also noticed a reduction in m6A levels (50%) in the dyn2 deletion mutant. What the function of Dyn2 is and whether it is conserved remains to be determined.

Our IP-MS experiments with Mum2 identified the complex as described in the manuscript, however we did not find evidence of orthologs of RBM15 and Hakai. More follow up work is needed using in vivo and in vitro assays are needed to determine how m6A by the yeast MTC is regulated.

P3 top: Although m6A is the most abundant internal methylation variant, it is far below the methylation levels of cap-adjacent nucleotides in mammalian mRNAs (PMID: 35970556 ).

We have added the word “internal” to the first sentence of the introduction.

A list of author contributions is missing.

We have added this in the revised version.

**Reviewer #2 (Recommendations For The Authors):**
Most of the conclusions of this paper are well supported by data, and the text is clearly written and easy to read. Here are my suggestions and comments:1. In Fig.2, why not use LC-MS to measure m6A levels in Ygl036w, Dyn2, Pab1, Npl3 mutants, as in Fig.1?

For measuring m6A levels, we use combination of LC-MS and m6A ELISA and m6A-seq2 throughout the manuscript. We used ELISA in the Fig2 because we had established this assay in the lab (Ensinck et al, RNA Journal, 2023). M6A-ELISA technique was more accessible and easier to execute compared to LC-MS. Additionally our collaborator for the LC-MS moved his lab to another country, which made it impractical to continue the use of LC-MS.

1. The protein purification experiment described in Fig. 4D is informative. Can they include Dyn2 in the expression system as well?

Thank you for the suggestion. Dyn2 was not the focus of the manuscript as Dyn2 has, at best, only a minor role in m6A deposition in vivo. We are also currently aiming to dissect how Dyn2 regulates m6A and the yeast MTC in follow up work. Hence we decided not to add more experiments on Dyn2 to the current manuscript.

1. Among the MTC components identified in this study, Dyn2 is a new and interesting subunit. It was shown that in *C. elegans* Dlc1 is involved in stabilizing the m6A writer Mett10. I wonder if yeast has a homolog of C. elegans Mett10?

As far as we know, there is no ortholog identified of Mett10 (METTL16 in mammals) in budding yeast.

1. The authors have emphasized "the m6A dependent and independent functions"; however, this is only based on previous observations. Is it possible that the less severe phenotype associated with ime4 catalytic mutant is due to residual catalytic activity? I think the data presented in Fig. 5 tell us that Ime4 and other MTC subunits have no additional moonlighting function. It is not entirely clear to me what "the m6A-independent function" is.

The observation that the yeast MTC complex has m6A dependent and independent function is based on the previous observations and the current work. In Agarwala et al 2012 PLOS Genetics, it was shown that mum2 and ime4 deletion mutants have more severe phenotype than slz1 deletion mutant or the catalytically inactive mutant of Ime4. We confirmed these observations in the revised manuscript (see Figure S5A and S5B). In this work, we showed that kar4 and vir1 deletion mutants have comparable delay in the onset of meiosis as mum2 and ime4 deletion mutants. Also, the MTC remains intact with absence of Slz1, but falls apart in ime4D, mum2D, vir1D or showed strongly reduced RNA binding (kar4 deletion mutant). Based on this we conclude that an m6A independent function of the MTC exists.

We have included data demonstrating that the catalytically inactive mutant has no residual m6A and a milder meiotic phenotype compared to the ime4 deletion mutant (see Figure S5A and S5B).

1. In Mum2-TEV-ProA IP (1B) and Kar4-TEV-ProA IP (S1A), Slz1 was not significantly enriched; however, in the repeated Mum2-TEV-ProA IP with/without RNAse (S1B, 4C), Slz1 was strongly enriched. Why are the Slz1 results so variable?

This is an astute observation, for which we do not have a definitive answer. One possibility is that Slz1 is the only subunit that is induced during meiosis. It is possible that induction of Slz1 varied between the different IP-MS experiments, hence leading to variability in its association with the MTC complex.

1. The last paragraph on page 11, "Collectively...", and the first paragraph on page 12, "Collectively...", seem redundant.

We have removed the duplicated paragraph in the revised manuscript.